# Disentangling climate and policy uncertainties for the Colorado River post-2026 operations

Bowen Wang [1,2] ✉, Benjamin Bass [1] ✉, Alex Hall[1], Stefan Rahimi [1,3] & Lei Huang[1]

Lakes Mead and Powell in the Colorado River Basin underpin water and hydropower supply for the western United States. While the policies currently regulating the basin will expire by 2026, planning remains challenging due to intertwined climate variability and policy uncertainties. Based on streamflow projections from 10 dynamically downscaled CMIP6 global climate models and unique methods that add and remove internal variability, we evaluate future conditions at Powell and Mead under existing and alternative policies. Due to projected streamflow declines, under existing policy, both reservoirs will face substantial risks (>80% likelihood) of reaching dead pool before 2060. Adopting recently proposed alternative policies reduces but doesn't eliminate such risks. All policies also exhibit tipping points where reservoir levels can change rapidly with a slight change in streamflow. A sustainable policy may require larger reductions to further reduce the reservoirs' dead pool risks and provide better buffers from sudden changes.

The Colorado River Basin (CRB) in the western United States (US) provides water and hydroelectricity for 35–40 million people in seven US states and Mexico primarily through its two major reservoirs, Lakes Powell and Mead[1,2]. Water supply operations in this critical basin are governed by a series of compacts, laws, and agreements, collectively known as the Law of the River[3]. This policy allocates a total of 15 million acre-feet (MAF; 18.50 km³) of water delivery to its recipient states and an additional 1.5 MAF (1.85 km³) to Mexico. However, the total amount of water delivered was agreed upon in 1922, based on the wettest sustained period in the CRB since 1520[4-7]. As water demand increases towards the full US allocation of 15 MAF (18.50 km³), this has led to an over-allocation of the river's water, posing serious challenges to the management of water resources across the western US.

The limited reliability of the system was highlighted at the end of the recent 2000–2022 megadrought in the southwestern US[8,9], which led to the lowest water levels in the basin's major reservoirs since their construction and the first federally declared water shortage for the CRB[10]. The megadrought was largely driven by a negative anomaly in precipitation associated with internal multi-decadal variability[8], although long-term anthropogenically-driven temperature increases exacerbated drought conditions in the CRB[11-13] and further reduced streamflow by roughly 10% during the drought[14]. Declines in the CRB's streamflow are likely to continue if greenhouse gas emissions are not curbed and temperatures continue to rise[15-19].

Understanding how climate change will affect the CRB and the functionality of the basin's major reservoirs is critically important, yet various sources of uncertainty make it challenging to confidently project future conditions. Projections of runoff in the CRB vary widely, given sensitivity to both the selection of Global Climate Models (GCMs)[17], which may display a wide range of future precipitation estimates[20], and interannual to decadal natural variabilities driven by large-scale circulations[21,22]. Furthermore, the sensitivity of runoff to changes in temperature and precipitation remains uncertain from observations and different hydrologic models[12,17,23-25], especially in the near-term when the spread in internal variability can exceed the expected changes due to climate change[26]. The emission trajectory

[1]Department of Atmospheric and Oceanic Sciences, University of California, Los Angeles, Los Angeles, CA, USA. [2]Department of Civil and Environmental Engineering, Massachusetts Institute of Technology, Cambridge, MA, USA. [3]Department of Atmospheric Science, University of Wyoming, Laramie, WY, USA. ✉e-mail: bowenwang23@g.ucla.edu; benb0228@g.ucla.edu

selected adds another layer of uncertainty regarding the level of warming projected in the CRB[17,27]. Finally, as current policies regulating water management of the CRB's major reservoirs are scheduled to expire by 2026[28], the policies that will regulate water delivery and consumption in the CRB are yet to be decided and remain highly uncertain[29,30]. To maintain sustainable water levels at Lakes Powell and Mead, the policy agreed upon will determine the amount of reductions in water delivery that should be imposed on both the Upper Basin (UB) and Lower Basin (LB; including Mexico, unless otherwise specified) states. Without proper management, the reservoirs may reach water levels that make them inoperable from hydropower and water supply perspectives, known as the inactive pool and the dead pool, respectively[31,32].

In this study, we developed a rapid and flexible Water Budget Model (WBM) that simulates reservoir operations in the CRB at an annual time step based on natural streamflow conditions in the Upper CRB (UCRB) as well as releases and evaporation at Lakes Powell and Mead (see "Methods"). Despite its simplicity relative to the models used by the Bureau of Reclamation (i.e., the Colorado River Simulation System, CRSS, and the Colorado River Mid-term Modeling System, CRMMS)[33], our model is able to accurately capture historical changes in storage at Lakes Powell and Mead (Figs. S1–5). While the model does not simulate how the reservoirs' storages change at sub-annual time-scales like CRSS or CRMMS, it is a more computationally efficient tool to help us understand the long-term risks in the Colorado River's water supply due to climate change in the 21st century.

Natural streamflow in the UCRB is the most important forcing variable to the WBM. While most recent CRB studies are based on climate projections made by GCMs from the 3rd and 5th phases of the Coupled Model Intercomparison Project (CMIP3, CMIP5)[17,32,34,35], we force our WBM with projected UCRB streamflow throughout the 21st century from dynamically downscaled, bias-corrected regional climate simulations based on 10 CMIP6 GCMs, combined with a calibrated hydrologic model[14,36–39]. Our projections are based on the SSP3-7.0 emission scenario, a business-as-usual trajectory of medium-to-high level of emissions, more realistic than the high-end SSP5-8.5 scenario[40]. The CMIP6 models represent the hydroclimate of UCRB more accurately than their CMIP3 and CMIP5 counterparts[41]. The high-resolution dynamical downscaling and calibrated hydrologic modeling - as opposed to a more commonly used statistical downscaling approach - provide a more realistic representation of the complex physical processes critical to projecting nonstationary future changes in the hydrology of UCRB. Based on our downscaled CMIP6 ensemble, we evaluate the reservoirs' future conditions using probabilistic methods that sample internal variability and a Decision Making under Deep Uncertainty (DMDU) approach that removes internal variability. These unique methods provide critical insight regarding the performance of the existing and alternative policies under a changing climate.

In the WBM, water management policies regulate reservoir releases based on the conditions of the reservoirs (primarily their water levels and storage). The policy determines the amount of water delivered to UB states, LB states, and Mexico and the release volume from Powell to Mead (see "Methods"). Our analysis considers four policy scenarios, including the existing policy based on the 2019 Colorado River Drought Contingency Plan (DCP2019) (Supplementary Tables S1, S2), a more stringent version of DCP2019 that increases water delivery reductions at each water level threshold (DCP2019 +), and two alternatives proposed by the UB (UB2024) and LB (LB2024) states, respectively, in March 2024 (see "Methods")[42,43]. The most important difference between the policies is that in DCP2019 and DCP2019 +, reductions in water delivery are decided based on the water levels of individual reservoirs, while LB2024 and UB2024 define delivery reduction thresholds based on the combined storage of Powell and Mead. Since none of the policies specifies an upper limit on UB water delivery, we use UB water demand projections made by the

Upper Colorado River Commission with an adjustment to match historically observed UB depletions (Supplementary Fig. S6)[44]. Among the 4 policy options considered, LB2024 is the only policy that requires reductions in UB consumption under drier conditions. Since UB2024 does not explicitly mention reductions for Mexico, we assume that reductions for Mexico are at the same proportions as the rest of the LB. We also note that LB2024 and UB2024 were proposed as the desired solutions of the LB and UB states, respectively, as of March 2024. Neither proposal represents the results of compromise among the stakeholders, nor is it likely to be officially adopted as the post-2026 management policy. In particular, it is unlikely for the UB water demand to be fully met without reductions. However, we choose to model these policies because they were the most recent, formally submitted proposals from the states that receive water from the Colorado River at the time of analysis. While we recognize that new alternatives are continuously being proposed[45], the alternatives assessed here reflect variations to existing policy that are currently considered for post-2026 operations (e.g., combined storage management approach and the general level of water delivery reductions).

We focus our analysis on the 2027–2060 period and the 2060–2100 period. The 2027–2060 period provides insight on the near-term, stakeholder-relevant planning period for the post-2026 operations decision-making, and the 2060–2100 period provides a longer-term perspective on the continued impact of climate change on the CRB. Our climate- and policy-driven analysis disentangles these two sources of uncertainty in the CRB's future and will help policymakers make informed decisions in sustainable water management based on state-of-the-art hydroclimate projections. Our analysis relies on a specific hydroclimate modeling workflow, which includes GCM selection, downscaling, bias correction, hydrologic modeling, and water budget modeling. Although modeling choices introduce uncertainty at each of these steps[46], each of these choices has been carefully validated.

This study aims to contribute to ongoing discussions on the future sustainability of the CRB system by evaluating its performance with existing and alternative policy options under projected climate conditions. Recent efforts have assessed the performance of Lakes Powell and Mead if recent megadrought conditions were to continue[1]. Here, we build upon previous efforts by assessing the CRB under state-of-the-art hydrologic projections and unique methodologies that both sample internal climate variability for probabilistic analysis and remove it to isolate system performance for a range of plausible natural flow conditions. These complementary sampling methods allow us to explore and elucidate the reliability of different policies under projected CRB hydroclimate conditions. Neither CMIP6 projections nor our sampling methods have previously been utilized by the Bureau of Reclamation, and they offer new and valuable insights into the management of the CRB's major reservoirs under existing conditions and the alternatives evaluated.

## Results
### Projected hydroclimate changes
In this paper, we focus on future hydroclimate changes across the UCRB, or the drainage area that flows into Lake Powell, since it encompasses the majority of the basin's flow and drives our WBM. While the 10 downscaled GCMs unanimously project large temperature increases in the UCRB (ensemble mean of 0.57 ± 0.10 °C/decade, Fig. 1A and Supplementray Table S3), there is little agreement on the sign of the long-term trends in precipitation over the UCRB, and the ensemble mean trend is close to neutral (−0.23 ± 0.49%/decade) (Fig. 1B). While precipitation dictates the patterns of streamflow variability within each GCM and uncertainty in the runoff trend across GCMs (Supplementary Fig. S7), all GCMs project a decreasing trend in runoff due to increased evapotranspiration from increased temperatures. The GCM ensemble-mean streamflow trend since 1984

(− 2.6 ± 1.2%/decade, Fig. 1C) equates to a temperature sensitivity of − 4.7%/°C relative to the ensemble-mean temperature projections of the CRB under the SSP3-7.0 scenario. This leads to an eventual decrease of 20.0% (± 9.3%) by 2060 and 30.5% (± 14.1%) in streamflow by 2100, which represents an intermediate estimate relative to previous studies[6,17,18,47]. If the existing policies were to remain in place for the rest of the century, the WBM predicts large declines in future water levels. The ensemble mean water level drops close to the inactive pool as early as the 2040s, and 9 of the 10 GCMs project that the Lake Mead water level by 2100 will be lower than the historically low condition in 2022 (Fig. 1D). In other words, despite large uncertainties and variabilities in projected precipitation and runoff, there is clear evidence that the existing policy is insufficient to sustain the reservoirs from the combined threats of temperature-driven declines in runoff and projected increases in UB water demand.

## Future reservoir conditions in the CRB

To evaluate the effect of changing water management policies with a robust representation of future hydrologic uncertainty due to internal variability, we develop a Monte Carlo simulation approach that expands from our 10-member dynamically downscaled GCM ensemble. We take the linear trend in streamflow from each GCM, which we assume to represent the anthropogenic climate change signal, and randomly sample and superimpose the residuals to account for the interannual variabilities (see "Methods", Supplementary Fig. S8). Starting from the 2022 conditions, for each downscaled GCM, we force the WBM with 1000 realizations of interannual variability imposed on the anthropogenic signal of that GCM to produce probabilistic estimates of the reservoirs' water levels under different policies. This provides an ensemble of 10,000 time series of UCRB runoff to force our WBM.

Figure 2 shows the probabilities of Lakes Mead and Powell reaching their respective inactive and dead pool levels from 2022 (the reservoirs' initial conditions) until 2100 under SSP 3–7.0. Across all policy scenarios, the risk of the two reservoirs reaching inactive or

dead pools increases from 2022 to 2100. This is expected given the declining runoff trend in all GCMs (Fig. 1C). Under DCP2019 (i.e., existing policy), both Powell and Mead will experience substantial risks of reaching critical conditions like inactive and dead pools. From the perspective of risks at individual years (Fig. 2A, D), 16% and 24% of all simulations project Powell and Mead, respectively, to be at the dead pool in the year 2060. This risk is elevated to 27% and 46% in 2100. However, if the goal is to ensure that the reservoirs never reach dead pools at all, the risk becomes much higher. Under DCP2019, Lakes Powell and Mead have an 85% and 83% chance, respectively, of reaching a dead pool at least once before 2060, and both have a 97% chance to reach a dead pool at least once in the 2060–2100 period (Fig. 2B, C, E, F).

Implementing any of the three alternatives can considerably reduce the risks at Mead, both in terms of the risk in any individual year or the cumulative risks. In terms of individual year risks, there is only a 5% chance that Mead will be at the dead pool in 2060 under LB2024, 7% under UB2024, and 11% under DCP2019 +. However, the disparity in the projected cumulative risks is much larger among the alternatives. LB2024 yields the lowest cumulative risk of 17% by 2060, while under DCP2019 + the risk is 65%, almost 4 times higher. Toward the end of the century, the risks faced by Lake Mead will substantially increase across all of the considered policies. Even under LB2024, the most effective policy in lowering risks at Mead, there is a 54% probability that Mead reaches a dead pool at least once in the 2060–2100 period, while the likelihood is 92% and 78% under DCP2019 + and UB2024, respectively. Regardless of the policy scenarios, the water level Lake Mead reached at the end of the 2000–2022 megadrought is likely to reoccur, with at least 65% probability of occurring at least once between 2027–2060 and at least 90% between 2060–2100.

For Lake Powell, while the risk of reaching dead and inactive pools is comparable to that of Mead under the alternative policies, the likelihood of reaching dead and inactive pools at least once remains relatively high. Before 2060, UB2024 generally yields both the lowest individual year risks (6%) and cumulative risk (45%) of reaching a dead

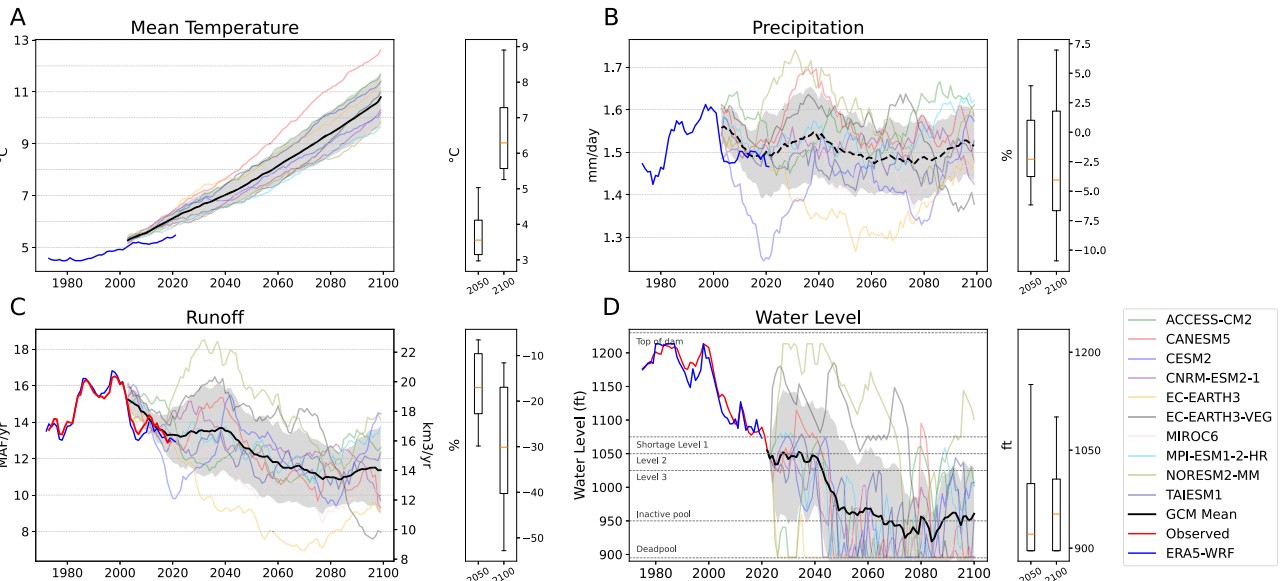

**Fig. 1 | Historical and projected future hydroclimatological conditions in the UCRB.** 20-year rolling mean of (**A**) temperature, (**B**) precipitation, (**C**) runoff in the UCRB, and (**D**) the projected Lake Mead water levels under existing policy. Gray shadings represent one standard deviation around the ensemble mean projections. The solid black lines in (**A**), (**C**), and (**D**) represent unanimous agreement on the sign of change, while the dashed black line in (**B**) represent a lack of two-thirds (i.e., at least 7 out of 10) GCM agreement on the sign of the future change. The box plots on the right of each panel show the range of projected changes in each variable by mid-

century and end-century from the 10 GCMs. Boxplot changes in (**A**), (**B**), and (**C**) are based on the linear trends of each GCM, while values in (**D**) are the projected water level values in the years 2050 and 2100. For the boxplots, the boxes represent the interquartile ranges (IQR). The solid orange line represents the median of the ensemble. The whiskers extend to the first/last data point whose distance to Q1/Q3 is within 1.5 times the IQR.

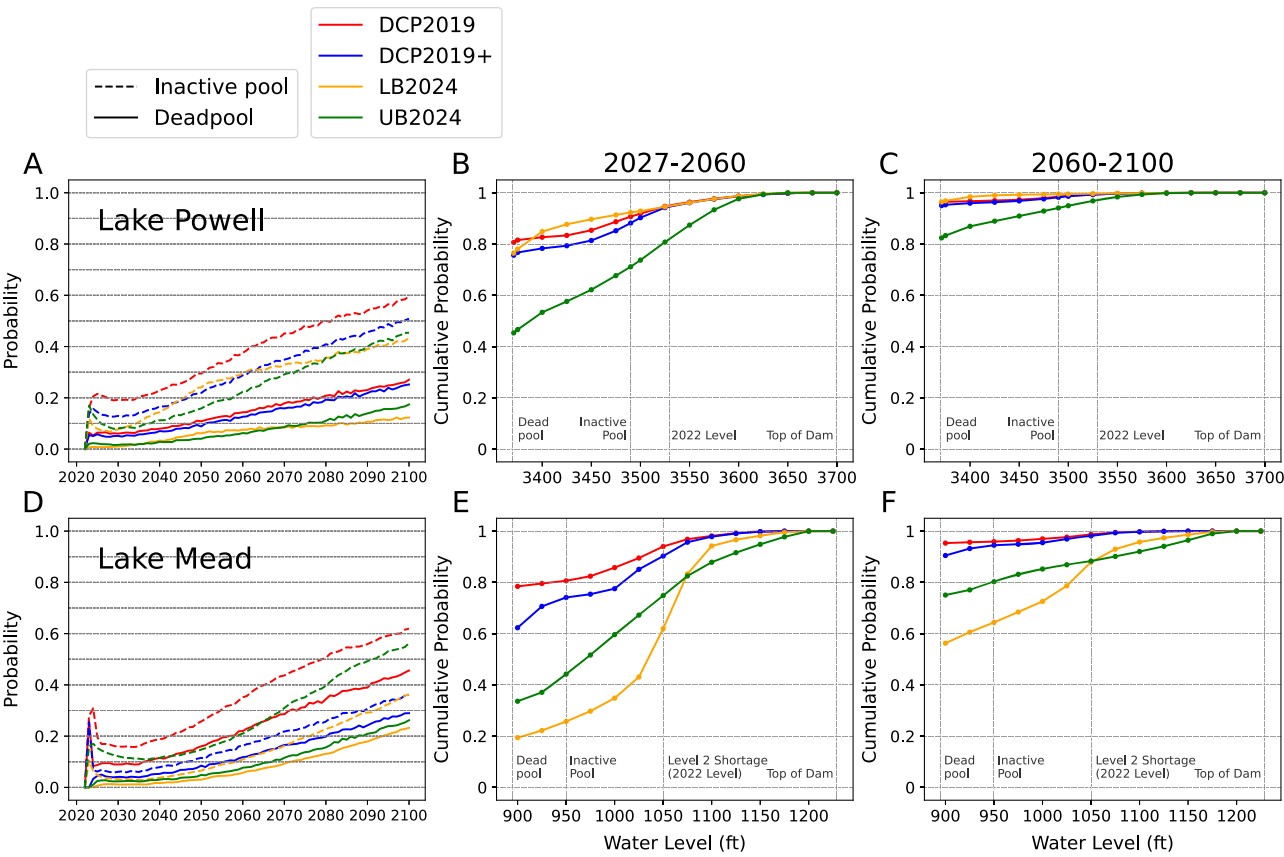

**Fig. 2 | Projected probabilistic risks at Lakes Powell and Mead in the 21st century. A** Probability of Lake Powell reaching its inactive and dead pool levels each year. **B, C** Cumulative probability of Lake Powell reaching particular water levels between (**B**) 2027–2060 and (**C**) 2060–2100. **D–F**, The same as (**A–C**), but for Lake Mead.

pool. It is the only policy with a projected cumulative risk lower than 75% (47%). Despite very similar individual year risk of reaching a dead pool to UB2024, LB2024 has a cumulative risk of 79%. In other words, in our ensemble, LB2024 is less capable of preventing Powell from reaching a dead pool but is better at recovering the reservoir content once a dead pool is reached and reducing the amount of time Powell spends as a dead pool. The cumulative dead pool risk under DCP2019 + is almost the same as under LB2024, but the individual year risks are considerably higher, indicating that DCP2019 + is generally ineffective from the perspective of reducing risks at Lake Powell. If we further consider the end-of-century conditions, it is likely that none of the alternatives can prevent Powell from becoming a dead pool at some point before 2100. It is almost certain (> 96%) that Powell will reach dead pool at least once between 2060 and 2100 under DCP2019 + or LB2024, and highly likely (83%) under UB2024.

**Evaluating impacts on water supply and hydropower generation**
The changing reservoir conditions at Powell and Mead due to the future climate and policy changes in the CRB can have major economic and environmental impacts in various possible aspects, including water quality, air quality, cultural/recreational resources, and biological resources[48,49]. Here, we focus on the two most critical aspects: water supply and hydropower generation[50,51]. Using the simulated results from our Monte Carlo ensemble, we calculate how the probabilities outlined in Fig. 2 translate to reductions in water deliveries (Fig. 3A, C) and hydropower (Fig. 3B, D, based on linear regressions from Supplementary Fig. S9). In both the 2027–2060 and 2060–2100 periods, DCP2019 is the policy that provides the greatest total water delivery to the LB and UB states combined. For example, for DCP2019, there is a 61% chance that the average total UB and LB delivery is at least 12 MAF/yr between 2027 and 2060. In comparison, the likelihood is about 54%

for DCP2019 +, 42% for LB2024, and 30% for UB2024. The additional water delivery comes at the expense of a higher risk of reaching inactive or dead pools due to lower water levels (Fig. 2) and subsequently lower hydropower generation (Fig. 3B, D). For example, under DCP2019, before 2060, there is a 54% chance that the two reservoirs can generate a total of 4.5 million MWh/yr, or 60% of the 2000-2010 mean. The likelihood increases to 67–82% under the alternatives. If we further consider the 2060–2100 period, DCP2019's advantage in water supply will diminish toward the end of the century (Fig. 3C), while its disadvantage in hydropower becomes even larger than in the 2027–2060 period. In other words, as streamflow continues to decline, implementing a policy that requires more reductions than DCP2019 will likely introduce considerable hydropower benefits with relatively small costs on water delivery.

To understand how water delivery and hydropower generation respond to different hydrologic conditions across policies, we partition the amount of water supply and hydropower generation received by the UB and LB states based on the mean UCRB natural flow occurring between 2027 and 2060 (Fig. 4). Since we explicitly control for the mean natural flow conditions, the main results here are insensitive to the choice of GCM and time period. The time period mainly affects the likelihood of each natural flow level occurring.

In terms of water supply, DCP2019, DCP2019 +, and UB2024 all promise to fully deliver the projected UB demand, which will increase from 4.3 MAF/yr in 2022 to 5.0 MAF/yr by 2060 (Supplementary Fig. S6), even in the driest conditions in our ensemble (- 8 MAF/yr natural flow). This also means that LB will receive the full impact of the decreasing natural flow. Under DCP2019, as outlined in the policy, the maximum reduction in delivery to the LB states is 1.375 MAF/yr, which will become the norm with a mean natural flow condition of 12.5 MAF/yr (Fig. 4A). However, if natural flow becomes even lower, delivery to

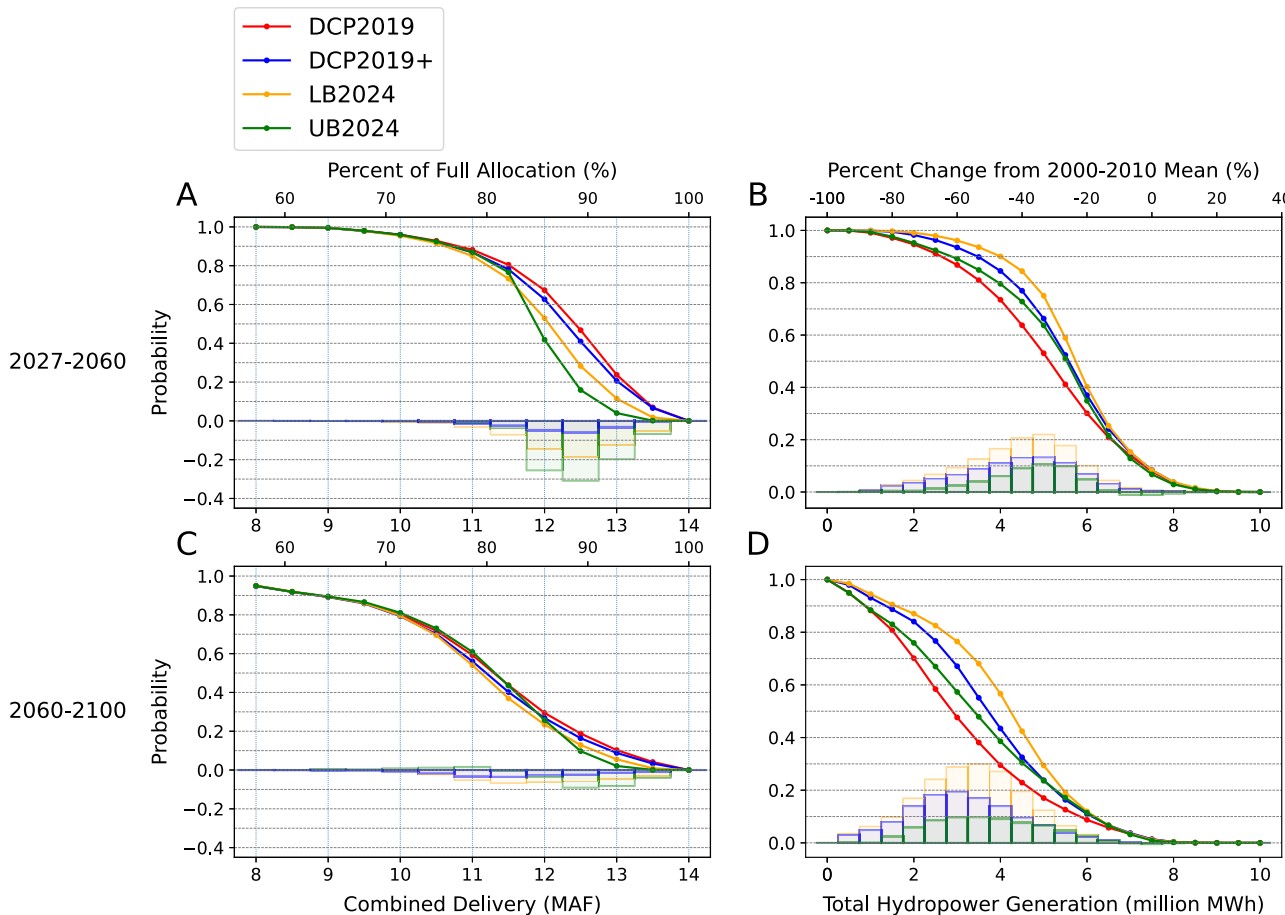

**Fig. 3 | Projected probabilistic water supply and hydropower risks in the 21st century.** Probabilities of the mean total (**A**, **C**) water delivery and (**B**, **D**) hydropower generation being below given levels between (**A**, **B**) 2027–2060 and (**C**, **D**) 2060–2100. The histograms represent the difference in probability between the existing policy DCP2019 and each of the alternatives.

the LB states will continue to decline at a roughly 1:1 rate relative to the decline in natural flow. Under such conditions, although not required by the policy, the delivery to LB states will have to decrease as the reservoirs are depleted. For this reason, the actual difference between DCP2019 and the alternatives is much smaller than the proposed delivery reductions (up to - 4 MAF/yr) may suggest. For example, when comparing DCP2019+ against DCP2019 (Fig. 4B), the actual difference in LB water delivery reaches a maximum of just 0.2 MAF/yr. UB2024 requires a much larger reduction to LB at wetter conditions around 14 MAF/yr mean natural flow (- 0.7MAF/yr), but practically makes very little difference from DCP2019 or DCP2019 + under dryer conditions (Fig. 4D). Unlike the three other alternatives, LB2024 is the only alternative that requires reductions to UB depletions (Fig. 4C). Although LB still takes much of the reductions at wetter conditions, LB2024 requires UB to take as much as a 1 MAF/yr reduction when natural flow is below - 10 MAF/yr, which LB receives.

In terms of the hydropower benefits, modest LB delivery reductions in DCP2019 + yields considerable amounts of additional hydropower, especially when the mean UCRB natural flow is 9–12 MAF/yr (Fig. 4F). Under DCP2019 +, the Glen Canyon Dam at Powell can at most generate an additional 0.6 million MWh/yr, while the Hoover Dam at Mead can generate an additional 0.3 million MWh/yr. LB2024 provides the largest hydropower benefits among all policy options (Fig. 4G). The peak hydropower gain occurs at - 11 MAF/yr for both reservoirs, producing an additional 0.6 and 0.9 million MWh/yr at Powell and Mead, respectively, the sum of which is about 20% of the mean generation between 2000 and 2010. UB2024 is able to generate the least amount of hydropower among the three alternatives, never

exceeding 0.35 and 0.25 MAF/yr at Powell and Mead, respectively, at any mean streamflow level (Fig. 4H). The positive changes occur when the streamflow levels are low and UB2024 makes no substantial change to the mean water deliveries. At the wetter conditions (> 14 MAF/yr) when UB2024 has the largest impact on water deliveries, the net combined impact on hydropower generation is neutral at best and often slightly negative.

## Equilibrium state of the reservoirs

We further address the hydrologic and policy uncertainties of the CRB using a GCM-informed DMDU approach[35,52–54], which outlines the reservoirs' water levels under a full range of plausible hydrologic future conditions. Unlike previous analyses in which we use GCM-derived streamflow either directly or through Monte Carlo simulations, here we force the Powell-Mead system with a constant volume of natural flow so that both reservoirs can reach their respective equilibrium water levels (Fig. 5)[54]. We perform this procedure with different constant inflow volumes that represent the range of the 1984–2100 runoff trend from all 10 GCMs (Supplementary Fig. S8), deriving the CRB system's equilibrium states across all plausible conditions. Unlike Figs. 2–4 that introduce internal variability, this approach removes climate variability to isolate the effect of each policy on the CRB's water system. We contextualize the results by relating the steady annual natural flow volumes to the years in which the natural flow conditions are expected to occur based on the linear trend of projected UCRB natural flow from our GCM ensemble (Fig. 5A), and the global warming levels associated with these years (Supplementary Fig. S10). To account for the time the reservoirs require to equilibrate,

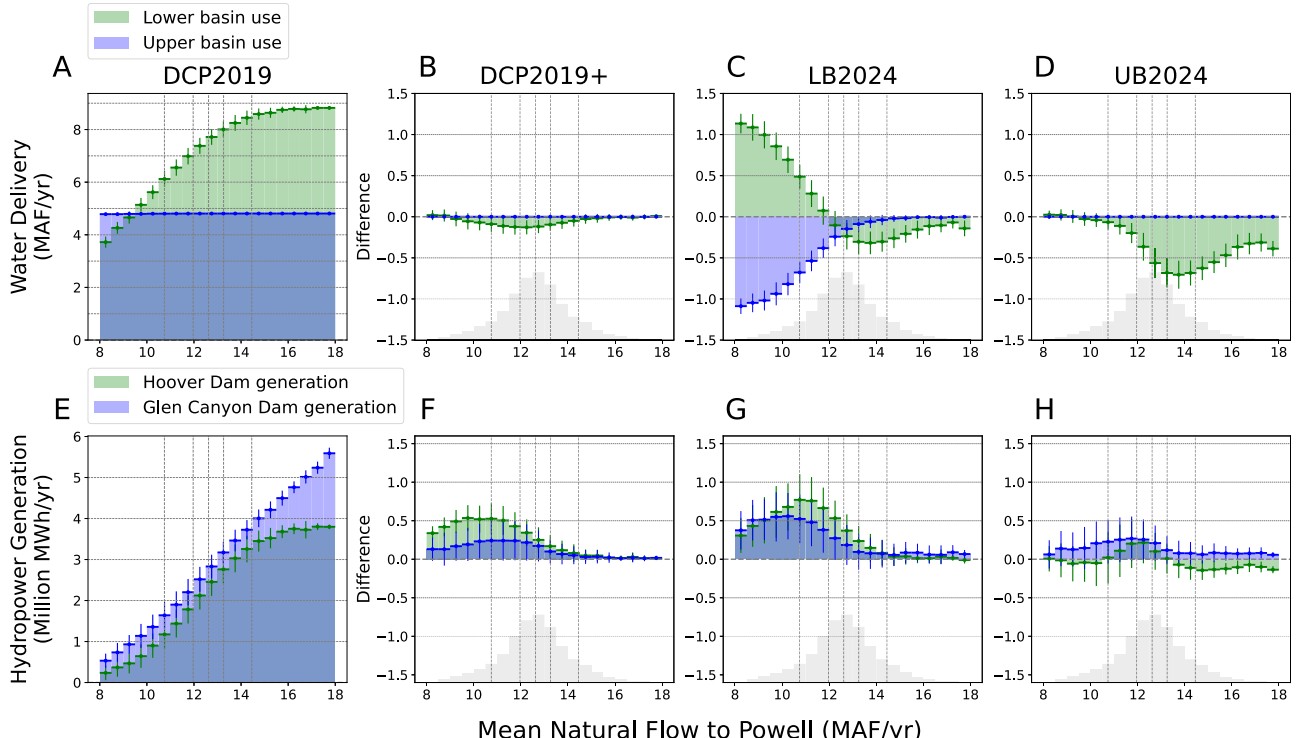

**Fig. 4 | Impact of policy and streamflow conditions on projected water supply and hydropower. A** Mean UB and LB deliveries under DCP2019 binned by mean UCRB natural flow between 2027–2060. **B–D** Mean difference in UB and LB depletions in 2027–2060 between DCP2019 and each of the 3 alternatives. Error bars represent the standard deviation in each bin. **E–H**, Same as (**A–D**) respectively, but for hydropower generated at the Glen Canyon (Powell) and Hoover Dams (Mead). The gray histograms show the likelihood of each mean natural flow level occurring between 2027 and 2060. The dashed vertical lines show the 10th, 30th, 50th, 70th, and 90th percentile mean natural flow in this period.

which typically occurs within 15 years (Supplementary Fig. S11), the results in Fig. 5B–G are matched to streamflow conditions averaged over the previous 15-year period. For example, the 2060 equilibrium state is reflective of the 2045–2060 average natural flow.

From 1984 to 2100, there is a 28.8% decrease in UCRB natural flow based on the bias-corrected GCM ensemble-mean linear trend (Fig. 5A). While it is expected that, regardless of the policy, total storage will decline as natural flow declines (Fig. 5B), the choice of policy affects the rate at which the equilibrium states are altered by changes in natural flow and the distribution of risk between the two reservoirs at a given natural flow. For example, under the existing DCP2019, both Powell and Mead will become dead pools when the CRB is forced with a steady 12.4 MAF/yr natural flow, which is expected by the GCM ensemble mean to occur in 2058. In contrast, under the enhanced DCP2019 +, the same 2060 natural flow translates to an equilibrium state of 13.2 MAF combined storage, and the two reservoirs will only be emptied when the natural flow is further reduced to 9.5 MAF/yr, which is expected to occur in 2131 if the GCM ensemble mean linear trend holds true beyond 2100. In terms of total storage, LB2024 can be seen as an enhanced version of DCP2019 +, but primarily through keeping storage higher at Mead between 10–13 MAF/yr. Under LB2024, Powell storage is similar to or even lower than that in DCP2019+ after a rapid drop occurs when natural flow is around 13 MAF/yr. UB2024 can be seen as the opposite of LB2024, such that it ensures a more gradual decline in the equilibrium Powell storage, while allowing Mead storage to drop rapidly from near full to empty at around 12.5 MAF/yr.

Notably, the equilibrium reservoir storage can drop rapidly with a modest amount of decrease in natural flow, forming tipping points in the reservoir system. For example, under DCP2019, the combined Powell and Mead storage can transition from full to empty as the natural flow decreases by what is expected to happen in just 45 years (2010 to 2055), primarily because the release from Mead is reduced at a much slower rate than the decline in natural flow. DCP2019 + generally exhibits similar tipping point patterns when natural flow is relatively high. However, it buffers the rapid drop when natural flow is below 13 MAF/yr by further reducing delivery to LB in these drier conditions. LB2024 and UB2024, in contrast, form distinct tipping point patterns at each reservoir. LB2024 produces a particularly steep drop in Powell storage such that the equilibrium reservoir storage can drop from full to empty when the natural flow declines by an amount that is expected to occur in less than 10 years. This is because in this brief period, Powell experiences a combination of declining natural flow, increasing UB depletions, and an increase in Powell release. UB2024 generates a similar pattern for Lake Mead when a decline in release from Powell (and thus inflow to Mead) is met with unchanging LB depletions, which quickly drains the reservoir.

In addition, surprisingly, depending on the policy, the equilibrium storage at each reservoir does not necessarily change monotonically. Most notably in LB2024, the equilibrium Mead storage drops from 25 to 7 MAF as natural flow declines from 14 to 13.2 MAF/yr (Fig. 5D). Storage then rises again to quickly reach full storage as the natural flow further declines to ~13 MAF/yr, while the equilibrium Powell storage plummets at the same time from full to empty. After this, Mead storage declines monotonically and more gradually, while Powell storage briefly rebounds to ~5 MAF before dropping to a dead pool again.

In terms of water supply, DCP2019, DCP2019 +, and UB2024 are all able to meet full UB demand in all the natural flow conditions considered here (Fig. 5E). For these policies, the LB delivery decreases roughly at a 1:1 rate as natural flow decreases (Fig. 5F). Under LB2024, in contrast, the reduction to UB depletion starts to take place when natural flow is just below 13 MAF/yr, reaching reductions of 1.3 MAF/yr by the 2100 conditions. As a result, under LB2024, LB states receive more delivery relative to the three other policies, which emerges when natural flow is below 12 MAF/yr (which is expected to occur in the

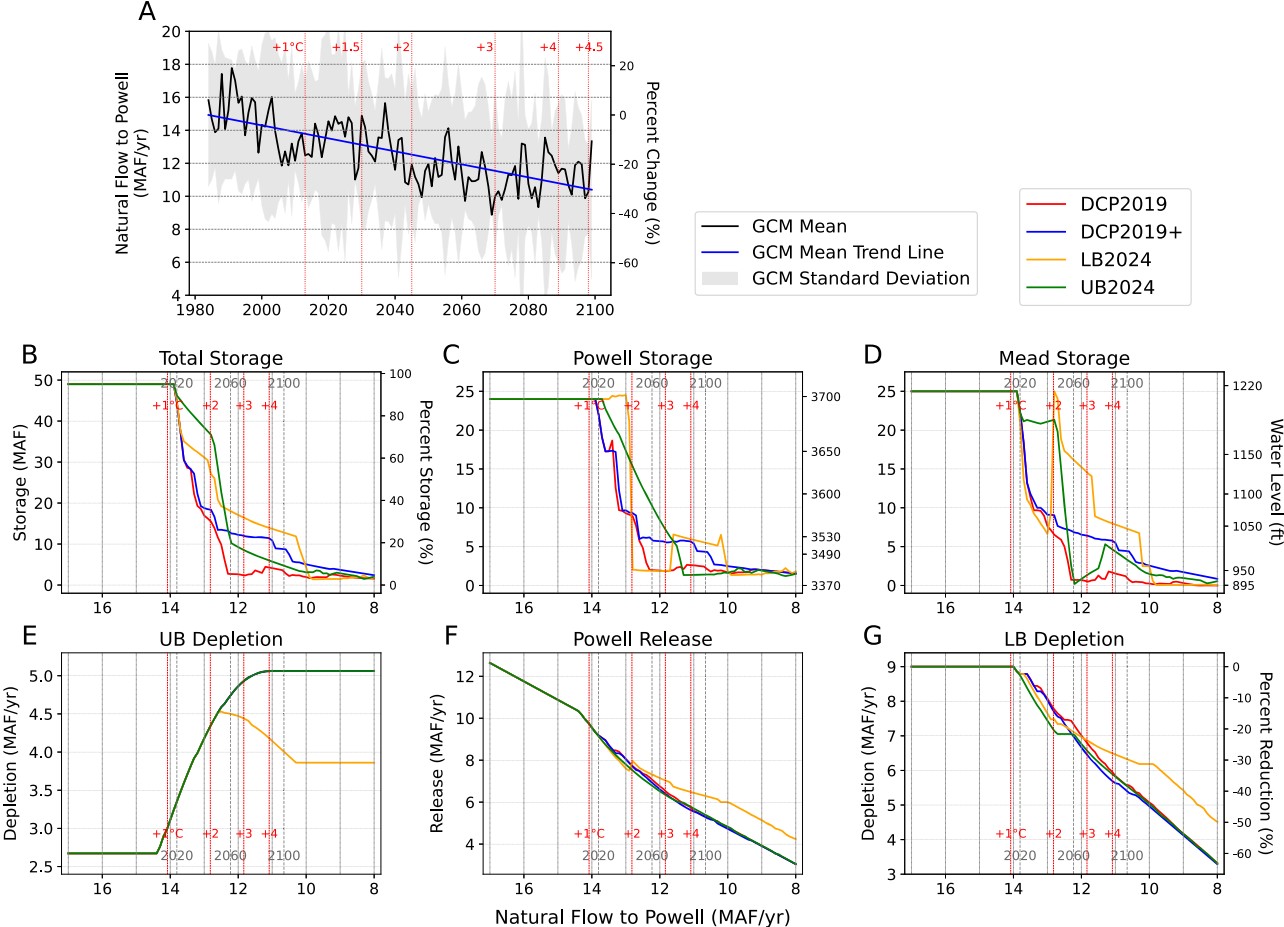

**Fig. 5 | Reservoir equilibrium states under a realistic range of streamflow conditions. A** GCM ensemble mean and standard deviation in the UCRB natural flow from 1984 to 2100 with linear trend fit to the ensemble mean (see "Methods", Supplementary Fig. S3a). **B**–**G** Equilibrium state conditions of (**B**) total Powell and Mead storage, (**C**, **D**) Powell and Mead storage/water level, (**E**) UB depletions, (**F**) Powell release, and (**G**) LB depletions under existing and alternative policy options. In (**B**–**G**), gray dashed lines represent the mean natural flow of the 15 years preceding 2020, 2060, and 2100, based on the linear relationship in (**A**). Red dashed lines represent the ensemble-mean linear trend natural flow corresponding to the year when different warming levels are reached. In (**E**), the curves for DCP2019, DCP2019 +, and UB2024 overlap completely. The global warming levels are derived from a 3rd-order polynomial fit, based on the ensemble mean of the 10 raw GCMs used in this study (Supplementary Fig. S10).

2060s), a threshold that is consistent with results in Fig. 4C. The LB depletion is buffered by the LB2024 policy and remains at 6.2 MAF/yr even when the annual natural flow is as low as 9.8 MAF/yr, below which LB depletion will decrease at the same rate as natural flow.

## Discussion

In this study, we evaluated the future conditions and risks faced by the CRB water management system under both existing and alternative policy conditions. Based on our ensemble of 10 dynamically downscaled CMIP6 GCMs, we developed a Monte Carlo approach that probabilistically considers internal variability, which allowed us to understand the likelihood of a water level being reached throughout the 21st century. We also evaluated how projected changes may impact the reliability of future water supply and hydropower generation. In addition, using our GCM-informed DMDU approach, we evaluated the equilibrium state of the reservoirs under a range of plausible natural flow conditions representative of future years or warming levels, without the influence of internal variability. These complementary approaches provide unique scientific and policy-oriented information critical to understanding how the CRB will respond to climate change.

We provided strong evidence that the CRB's reservoirs and water supply reliability will experience high risks in the coming decades if the current policy is to remain in place. This is true whether we use the streamflow projections from our 10-member dynamically downscaled GCM ensemble, the enlarged Monte Carlo ensemble that resamples internal variability, or the DMDU approach that removes the influence of internal variability. Without a change in policy, it shall not be a surprise if both reservoirs reach dead pools at some point before 2060, and the LB states will bear essentially all water supply costs of reduced natural streamflow in the basin, while the UB water demand will continue to be met under virtually all plausible conditions during the 2027–2060 planning period (Fig. 4A).

We tested the performance of three alternative policies that outline water delivery reductions to LB and/or UB water users, including one policy reflective of the existing policy, albeit with greater water cuts (DCP2019 +), and two that represent the most recent proposals by the LB (LB2024) and UB states (UB2024). Based on our Monte Carlo simulations, all three alternatives can substantially reduce future risks at Lake Mead and will leave Lake Powell at a similar level of vulnerability as under DCP2019 (Fig. 2). However, the three alternatives offer very different pathways for managing the reservoirs' risks. DCP2019 + provides the most modest change to the current structure of water supply deliveries (Fig. 3A), while providing moderate hydropower benefits to both reservoirs, especially under drier natural flow conditions (Fig. 4F). Despite the seemingly large maximum reduction rate

(> 4 MAF/yr) when Mead is at its lowest water levels, the DCP2019 + effect on mean reductions over time is much less than 4 MAF/yr since this policy is able to raise the reservoir's water level back to a normal range much faster than DCP2019.

UB2024 and LB2024 require a similar level of overall water delivery reductions (Fig. 3) but differ on how to distribute them. UB2024 keeps the UB demand fully satisfied and assigns large reductions to the LB delivery. It leads to the lowest risk at Powell among all alternatives, but is not as effective as LB2024 in terms of reducing risk at Mead. When compared to DCP2019, UB2024 effectively only reduces deliveries at wetter conditions. In the drier conditions when management is most needed, UB2024 is practically the same as DCP2019. Although UB2024 may help generate slightly more hydropower than DCP2019 at Lake Powell, it is the only alternative that creates a net hydropower loss under any mean streamflow condition. In contrast, LB2024 is the only policy evaluated here that proposes reductions to UB depletions, thereby allowing for higher LB deliveries relative to the other alternatives, especially under drier conditions. LB2024 yields the lowest risks at Mead and generally the highest hydropower output under any hydrologic condition, but this policy relegates higher risks to Powell. Although LB2024 is relatively successful at lowering risks in the 2027–2060 period, the risk of facing low reservoir water levels and substantially reducing CRB water supply remain high toward the end of the century (Fig. 3C). This demonstrates the temporary nature of the post-2026 alternatives and the need to design and adopt new policies towards the end of the century if greenhouse gas emissions continue to increase and climate change further reduces the amount of water available to the CRB.

With the DMDU analysis, we were able to isolate policy characteristics from climate variability across a range of plausible, steady natural flow conditions. The general patterns match well with our results in the probabilistic analysis in terms of both the overall risks and impacts to each reservoir. This idealized experiment shows critical insights that would have been otherwise neglected in the probabilistic approach. First, it highlights the tipping points in the CRB system and demonstrates how reservoir storage can decline dramatically due to small decreases in natural flow, transitioning from seemingly stable conditions to critical levels like inactive and dead pools. The presence of tipping points is prevalent across all policies, especially when reservoir release either stays unchanged or even increases with declining natural flows.

Second, we show that the reservoirs' equilibrium states do not always respond monotonically to changes in natural flow and may experience swings in storage conditions. This is especially notable in LB2024 and UB2024, which are governed by combined storage rather than water levels in the individual reservoirs. In these two alternatives, combined storage always declines when the constant natural flow declines, but the storage in each individual reservoir can fluctuate. For example, under LB2024, Mead storage would sharply decrease before rising again to full storage in a tight window of just ~1 MAF/yr. Although this pattern is not visible in the averaged numbers of the Monte Carlo simulations, diluted by realistic interannual variations in streamflow, these swings from equilibrium conditions show unintended incoherences within each policy, where the system response does not align with the hydrologic conditions. A more robust policy should ensure that sustained periods of lower inflow will lead to a reduction in downstream releases to preserve reservoir storage. In all of the policies evaluated here, there are some ranges of reservoir levels where water supply reductions are held the same. These are also the ranges where the tipping points occur. This suggests it is almost always better to have a scheme that linearly increases reductions in response to reservoir storage levels. Determining the exact reductions at each step is a subject for future work.

As the existing water policies are scheduled to expire by 2026, our study provides a timely analysis that combines state-of-the-art regional hydroclimate projections and recent policy proposals. We elucidate the risks and performance of the CRB water management system under existing and alternative policies via probabilistic and decision-making under uncertainty perspectives. We demonstrate a clear need to alleviate the climate-change-driven risks the CRB system will likely endure under the existing policy and the benefits of replacing it with alternative policies. We show how the alternatives proposed by the two major political entities involved, the UB and LB states, can alleviate risks but show notable disagreements on the future allocation of the basin's water supply and hydropower resources. We argue that a robust alternative policy should serve as a buffer to sudden changes in reservoir conditions and yield reservoir storage and water delivery outcomes consistent with the hydrologic conditions. In future work, as new policies are developed, they can be rapidly tested through our modeling framework along with our methods for introducing and removing internal variability. Even after such a policy can be agreed upon, in the longer future, CRB water management negotiations will likely be a recurring norm in response to ongoing climate change.

## Methods

### CRB and its Hydroclimate Projections

The Colorado River Basin (CRB) is a major source of water supply and hydropower for the western US through its two major reservoirs, Lake Powell and Lake Mead. It covers around 650,000 square kilometers (or 250,000 square miles) and is divided into the Upper and Lower CRB (UCRB, LCRB) (Supplementary Fig. S1). Lake Powell is located at the outlet of the UCRB, and Lake Mead is located in the upper reaches of the LCRB, downstream of Lake Powell. Although the UCRB and LCRB are similar in their surface area, streamflow in the UCRB contributes over 90% of the total streamflow in the CRB, and as further elaborated in the following sections, the UCRB streamflow is also what drives our WBM. Notably, the UCRB is officially defined to end by the Lees Ferry gauge, which is downstream of Lake Powell and receives runoff from one additional tributary, but for our purposes, we use the streamflow into Lake Powell and the UCRB streamflow interchangeably.

### Data

We used 9 km-resolution streamflow data simulated by the Noah-Multiparameterization (Noah-MP) land surface model forced by dynamically downscaled precipitation and temperature. Noah-MP was first forced by the fifth generation ECMWF Reanalysis (ERA5) dynamically downscaled by the Weather Research and Forecasting (WRF) model to 9 km resolution (ERA5-WRF)[36] and calibrated to achieve a realistic representation of streamflow conditions in the western US, and particularly, for the interest of this study, in the CRB[39]. Figure 1C shows the comparison between the observed and Noah-MP-simulated natural streamflow from the Upper Colorado River Basin into Lake Powell, which drives the Water Budget Model used to represent the reservoirs' water levels. The calibrated Noah-MP was then forced by 10 unique CMIP6 GCMs under the SSP3-7.0 scenario (Supplementary Table S3)[40]. These 10 GCMs were selected primarily based on their ability to accurately represent the hydroclimate of the western US, but to better sample model uncertainty, we also include a couple of weaker performing GCMs[55]. These 10 GCMs were dynamically downscaled using WRF[37] and subsequently bias-corrected using PresRat[56]. In this way, the projections from all dynamically downscaled GCMs are based on the same historical climatology. Streamflow data simulated from each of the GCMs were additionally bias-corrected using a simple mean state scaling to the ERA5-WRF streamflow data in the historical period (1984–2014). The scaling factors for all GCMs are within the range of ±10%.

### Water budget model (WBM)

The water budget model (WBM) translates the natural streamflow conditions from the UCRB to the managed reservoir conditions at

Lakes Powell and Mead (see Supplementary Fig. S2). The water budget accounts for inflow, outflow/release, and lake evaporation in calculating the volume of storage in each reservoir:

$$\text{Storage}_{\text{year}} = \text{Storage}_{\text{year}-1} + \text{Regulated Inflow}_{\text{year}} \\ - \text{Release}_{\text{year}} - \text{Evaporation}_{\text{year}} \quad (1)$$

The model operates on calendar year time steps because it is the time unit of water accounting for all the water deliveries/consumptions in the UB and LB states. Based on historically observed relationships, the WBM first uses linear regression to translate the volume of annual natural streamflow in UCRB to the volume of regulated inflow to Lake Powell (Supplementary Fig. S2a). The linear relationship between natural and regulated flows for year i is as follows:

$$\text{Powell regulated inflow}_i = (\text{natural flow}_i - \text{UB depletion}_i) * \alpha_1 + \beta_1 \quad (2)$$

where $\alpha_1 = 0.88$ and $\beta_1 = 1.22$, as determined by the relationship between observed annual regulated inflow, observed annual natural flow, and observed UB depletions in the historical period. Since UB depletions are explicitly accounted for, the linear regression can be understood as an implicit, statistical representation of the reservoir operations upstream of Lake Powell. During years of greater UCRB natural flow volumes, a part of the streamflow is conserved in upstream reservoirs, which will increase the inflow to Powell during years of low natural flow. While this linear regression implicitly assumes stationary operations by the upstream reservoirs, we have also tested an alternate version of the model with no regression that essentially assumes no upstream reservoirs:

$$\text{Powell regulated inflow}_{\text{year}} = \text{natural flow}_{\text{year}} - \text{UB depletion}_{\text{year}} \quad (3)$$

The projection results are generally insensitive to the choice between Eqs. (2) and (3) (Supplementary Fig. S12). However, (3) is arguably less realistic when the natural flow conditions are not dry enough to exhaust the upstream reservoirs, especially in the short term, which is why we show results based on (2) in the main text. In addition, this suggests that this linear regression translating natural flow to regulated flow is not central to the conclusions in this study, so the benefit of developing a non-stationary representation of the upstream reservoirs behaviors is expected to be minor.

We assume that future UB demand will grow in a similar pattern as the projections made by the Upper Colorado River Commission between 2016 and 2070[44]. Note that we excluded the evaporation estimates from the UCRC projections and made our own lake evaporation projections based on the changing reservoir conditions. We find that in the period when the UCRC projections of UB demand overlap with the historical period, assuming that historical UB demand has been fully met in the historical period and is equivalent to historical UB depletions, the UCRC projections have overestimated the UB demand by about 1 MAF/yr (Supplementary Fig. S6). For this reason, we adjusted the UCRC-projected UB demand by shifting all projections down by 1 MAF/yr and fitted the adjusted projections to a 3rd-order polynomial, which also largely matches with the historical growth of UB demand since 1960. In this way, we can estimate the UB demand for any individual year between 1960 and 2070. Since the UCRC projections end in 2070, for the years beyond 2070, we assume that UB demand will stay constant at the 2070 level.

The release from Powell each year is determined by the Interim Guidelines of 2007 (i.e., existing policy; IG2007; Supplementary Table S1) in the historical period and either IG2007 or one of the operating rules proposed by UB and LB states for future projections[42,43]. Each policy decides the release volume based on the present-year storage or water level at Powell and/or Mead. All the policies require that the Powell release makes sure the reservoir does not go over the reservoir's full capacity, given the present-year storage, inflow, and evaporation. Under IG2007 and LB2024, when the policy requires a balancing/equalization between Powell and Mead, the Powell release is the value such that the storage in the two reservoirs will be the same in the next year. Under LB2024, when Powell and Mead's combined storage is between 30% and 80% their full capacity, the release is determined by the mean UB depletion in the 3 years leading up to the present year. Otherwise, the release is either a predetermined number or calculated by a linear function given the reservoir's present-year storage. When the Powell level is below the inactive pool, its release is constrained by the physical limits of the river outlets outlined by Schmidt et al.[29].

We approximate the vast majority (99%) of inflow using the observed streamflow gauge along the Colorado River at Diamond Creek (USGS site number 09404200). The streamflow from this gauge has a strong correlation with the release from Powell and is thus represented with another linear regression to determine regulated inflow into Lake Mead (Supplementary Fig. S3b):

$$\text{Mead inflow}_{\text{year}} = \text{Powell release}_{\text{year}} * \alpha_2 + \beta_2 \quad (4)$$

where $\alpha_2 = 1.0001$ and $\beta_2 = 0.857$. Since $\alpha_2$ is almost identical to 1, we simplify this relationship by removing $\alpha_2$ and estimating the inflow to Mead as following:

$$\text{Mead inflow}_{\text{year}} = \text{Powell release}_{\text{year}} + \beta_2 \quad (5)$$

The physical meaning of $\beta_2$ is the amount of streamflow into the Colorado River in the section between Powell and Mead. The inter-annual variability and climate-driven changes in this section of the CRB can certainly affect the CRB's reservoir content and water deliveries. Here, we assume $\beta_2$ to be a constant to focus on the effect of UCRB streamflow changes on the CRB's future water supply. The observational streamflow gauge record in this section of the basin is also too short and sparse to provide a robust dataset to validate against.

With a few years of exception, releases from Mead closely resemble the combined LB consumptive use (i.e., California, Nevada, Arizona, and Mexico) recorded in the Water Accounting Reports, so we simply assume the following:

$$\text{Mead release}_{\text{year}} = \text{LB Consumptive Use}_{\text{year}} \quad (6)$$

Note that for the deliveries to Mexico, the consumptive use also includes the deliveries in excess of treaty requirements, which are low in most years but can be large in years of very high natural flow input to Powell. In addition, we find that since around 2003, the annual releases from Mead start to become consistently higher than the LB consumptive use. While there appears to be an increasing trend in the difference between Mead release and LB consumptive use, the record is too short for the trend to be conclusive, so for years after 2003 (thus including the entire future projection period), we calculate the Mead release by adding the average difference between Mead release and LB consumptive use to each year's annual LB consumptive use between 2003 and 2019 (i.e., 0.4 MAF/yr), which roughly represents the evaporative losses in the LB downstream of Lake Mead:

$$\text{Mead release}_{\text{year}} = \text{LB Consumptive Use}_{\text{year}} + 0.4\,\text{MAF/yr} \,(\text{year} \geq 2003) \quad (7)$$

Future annual LB consumptive use is by default set to be 9 MAF (11.1 km$^3$), which is the sum of the LB allocations. This does not account for the 1 MAF/yr of additional consumptive use that the original 1922 Colorado River Compact allows the LB states. This consumptive use is lowered for existing and alternative policies when the reservoirs'

content decreases. The existing policy DCP2019 and one of the alternatives DCP2019+ determine the reduction in LB delivery each year based on Lake Mead's water level. Reductions start to occur when water levels reach below the threshold of 1090 ft (332.2 m) by 0.241 MAF (0.30 km³). The maximum possible reduction under DCP2019 is 1.375 MAF (1.70 km³), which occurs once Lake Mead's water level reaches below 1025 ft. (312.4 m), while under DCP2019 +, the maximum reduction is 4.275 MAF when the Mead water level is below 950 ft. (289.6 m). Reductions at each threshold are detailed in Supplementary Table S3. For DCP2019 +, the LB reductions are based on the Action Alternative 1- 2025 policy described by the Bureau of Reclamation in the Near-term Colorado River Operations Draft Supplemental Environmental Impact Statement[48]. Operations at Lake Powell under DCP2019 + is based on IG2007, the same as DCP2019. The other two alternatives, LB2024 and UB2024, were proposed by the LB and UB states, respectively, in March[42,43]. The main difference between these two alternatives and DCP2019/DCP2019 + is that they define operations and delivery reductions primarily based on the combined reservoir storage in Powell and Mead. Note that UB2024 does not explicitly define the reductions by Mexico, which we assume to be the same proportion as the rest of the LB.

Evaporation from both reservoirs is determined by estimating the surface area of the reservoir using a 4th-order polynomial that represents the historically observed relationship between surface area and water level. The surface area of each reservoir is multiplied by a constant, estimated by Holman et al.[57] for Powell and Earp and Moreo[58] for Mead:

$$\text{Evaporation}_{\text{Powell}} \, [\text{acft}/\text{yr}] = 5.7 [\text{ft}/\text{yr}] * \text{Surface Area}_{\text{Powell}} \, [\text{ac}] \quad (8)$$

and

$$\text{Evaporation}_{\text{Mead}} \, [\text{acft}/\text{yr}] = 6.2 [\text{ft}/\text{yr}] * \text{Surface Area}_{\text{Mead}} \, [\text{ac}] \quad (9)$$

The effect of the warming climate on the evaporation rate at the reservoirs is minor compared to the effect of the changing surface areas and has a minor impact on the resulting water levels (Supplementary Fig. S13).

Supplementary Fig. S5 provides an evaluation of the WBM from 1975 to 2020 using the ERA5-WRF streamflow data, showing that the model is able to reproduce both the variability and trends in water levels over the historical period with reasonable accuracy (MAE = 11.2 ft for Powell and MAE = 10.9 ft for Lake Mead). It is important to note that although the WBM-predicted water levels may deviate from the observed values (e.g., early 2000s at Lake Powell), the model's explicit representation of interacting management policies between the reservoirs allows the model to self-adjust. For example, if the predicted streamflow is too low in certain years, that leads to underestimating water levels (as in the early 2000s), the reservoirs will respond by also releasing less water than observed, which will offset the streamflow underestimation. Vice versa, an overestimate of streamflow will make the reservoirs release more.

Our WBM is beneficial to our study because it is (1) free (Python-based), (2) computationally efficient, and (3) flexible with different policy alternatives. The Colorado River Simulation System (CRSS) requires a RiverWare license, and while it includes finer details like the diversions and smaller reservoirs in the CRB, its longer run-times make it infeasible to simulate over a large number of time series for long time-periods. In contrast, while our model does not explicitly represent these minor systems, it captures the cumulative impact on the total volume of water reaching Lakes Powell and Mead and reduces the amount of time needed for each simulation. Similarly, although CRSS can model local conservation efforts, our approach focuses on the aggregate effects of conservation or demand across the CRB. Further details on CRSS can be found in Bureau of Reclamation[33].

## Monte carlo simulation

Figure 1D shows the water level changes from 2022 to 2100 using the streamflow time series from our 10 dynamically downscaled GCMs. Due to the relatively limited sample size in our GCM ensemble, the projections are susceptible to internal variability. To more robustly represent the influence of random internal variability on the reservoirs' water levels, we enlarge our GCM ensemble using a Monte Carlo simulation. We first linearly detrend the natural streamflow from each of the GCMs' time series (Supplementary Fig. S6a). For each GCM, we then apply an autoregressive model of order 1, or AR(1), on the anomalies and separate the portions of the anomalies explained and unexplained by the autocorrelation. The fraction of the anomalies explained by the autocorrelation is the slope of the AR(1) model and represents the 1-year memory of natural flow variability. We divide the anomalies unexplained by AR(1) by the expected natural flow volume of that year based on the given GCM's natural flow linear trend, resulting in a time series of fractional anomalies. We then fit a Pearson-III distribution, a commonly used distribution for streamflow estimations, to the annual fractional anomalies to represent their probability distribution function (Supplementary Fig. S6b)[59–61]. Using other distribution types, like a lognormal distribution, makes a negligible impact on the results. In addition, the derived distributions for all GCMs are very similar, showing that this approach is generally insensitive to model choice and that the internal variability sampling approach is robust.

To create the ensemble, for each GCM, we first randomly sample a fractional anomaly for the first year (i.e., 2023) from the corresponding probability distribution function and add it onto the 2022 value of the GCM linear trend. That is, for year = 2023,

$$\text{Storage}_{\text{year}} = \text{Storage}_{\text{year}-1} + \text{Regulated Inflow}_{\text{year}} - \text{Release}_{\text{year}} - \text{Evaporation}_{\text{year}} \quad (10)$$

Starting from the next year, in addition to what we do for the first year, we account for autocorrelation by multiplying the strength of the memory and the anomaly from the last year, and add the product to get the natural flow for this year. In equation form, for year > 2023:

$$\begin{aligned}\text{Natural Flow}_{\text{year, GCM}} &= \text{Linear Trend Natural Flow}_{\text{year, GCM}} \\ &* (1 + \text{frac. anomaly}_{\text{year}}) + \text{memory}_{\text{GCM}} \\ &* \text{anomaly}_{\text{year}-1}\end{aligned} \quad (11)$$

where memory refers to the slope of the autoregressive model for each GCM. Also note that we are using the absolute anomaly value, rather than the fractions, since the change in the linear trend of natural flow in one year is negligible for practical purposes. Since the fractional anomalies are always greater than −1, this method ensures that the natural flow values generated are always positive. It also accounts for the declining trend in natural flow. For example, when the expected flow is lower, the sampled anomaly is less likely to be very large (either negative or positive) than when the expected flow from the linear trend is higher.

We performed 1000 such simulations from 2022–2100 per GCM per policy option, making an ensemble of 10,000 simulations per policy that were used as input for the WBM's probabilistic analysis. Each simulation starts from the Powell and Mead storages in 2022 as initial conditions.

Supplementary Fig. S14 compares our CMIP6-based hydrology ensemble with the ensembles developed by the Bureau of Reclamation in the 2027–2056 planning period[62]. Overall, the different ensembles are quite similar in their distributions, with the ensemble means near 13 MAF/yr. Our CMIP-based ensemble tends to be drier than the Reclamation's CMIP3 and CMIP5 ensembles but is more closely aligned with the non-CMIP-based ensemble, which statistically samples conditions

like the historical period and the paleoclimate 1576–1600 megadrought[32,63]. In terms of the extremes, our CMIP6 ensemble generates a wider range than any of the Reclamation ensembles. The wetter members in our CMIP6-based ensemble are comparable to their CMIP3 ensemble, with higher extremes likely due to the larger number of times we randomly sample internal variability. However, the drier end of the spectrum is likely not a result of chance. One of our dynamically downscaled GCMs (i.e., EC-Earth3) demonstrates much drier conditions in the 2027–2056 period than any of the conditions sampled by the Reclamation ensembles, thus producing a considerable number of members in our ensemble that represent a possible future hydrology drier than any of the Reclamation ensembles. It is also possible that our hydrology model (Noah-MP) and the hydrology models used to obtain the Reclamation's CMIP-3 and CMIP-5 ensemble of natural flow exhibit unique runoff sensitivities to changes in temperature and precipitation.

### Reservoir equilibrium states

To understand the responses of the Powell-Mead reservoir system to different streamflow conditions and the effect of policy changes irrespective of internal variability, we forced the WBM with constant amounts of inflow every year. When the constant inflow volume persists for a sufficient amount of time (typically within 15 years), the water levels of both reservoirs eventually stabilize and reach an equilibrium state or water level. We simulated the WBM with a range of realistic annual inflow volumes at 0.1 MAF (0.12 km$^3$) intervals under each of the four policy options. The simulation period lasted for 500 years, which is more than enough for the reservoirs to equilibrate. We took the mean of the storage at each reservoir and releases to both LB and UB states in the last 100 years of the 500-year simulation period to represent the equilibrium state of the reservoirs. We started the simulation at the 2022 levels, but as a result of the long simulation period, the equilibrium results are insensitive to the initial reservoir conditions. The resulting equilibrium states are thus properties of each reservoir under the specific conditions of the corresponding inflow volume and policy option.

We determine the years corresponding to each constant natural flow volume based on the linear trend fit to the GCM-ensemble mean runoff, with the 15-year period needed to equilibrate the reservoirs taken into account (Supplementary Fig. S11). The corresponding years then determine the UB demand in each equilibrium simulation. For example, a natural flow condition of 12.3 MAF/yr, based on the GCM-ensemble linear trend, represents the expected condition in the year 2050, or in other words, the average condition between the 15-year period of 2043–2058. This means that the constant inflow of 12.3 MAF/yr represents the equilibrium conditions of year 2058. The value of UB demand used in this simulation with a constant natural flow of 12.3 MAF/yr is then based on the mean UB demand between 2043 and 2058 based on the non-linear relationship in Supplementary Fig. S5.

This is a similar method to what is used in Wang and Rosenberg[54], who also fed constant volumes of inflow to a simple CRB water management model under different policy conditions. They looked at the time needed for the Powell and Mead to reach a certain storage level and the combined storage reached by the end of the planning period (i.e., 2060) under a small number of combinations of natural flow and policy conditions. The main difference between their implementation and ours is that we considered a continuous spectrum of plausible natural flow conditions, demonstrated how the equilibrium states change with the constant volume of natural flow input, and additionally contextualized the results based on our ensemble mean trend of the expected time for a natural flow condition to occur.

### Data availability

The dynamically downscaled ERA5 and GCM meteorological forcing data are available at https://registry.opendata.aws/wrf-cmip6/. Within this same AWS bucket, the bias-corrected forcing data is available at https://wrf-cmip6-noversioning.s3.amazonaws.com/index.html#ben_temp/d02_9km/FORCING/GCMs/Post_BC/ and the hydrology simulations are available at https://wrf-cmip6-noversioning.s3.amazonaws.com/index.html#ben_temp/d02_9km/Sims_LSM_Only/0_Final_post_BC/. The observed natural streamflow data for the Colorado Basin is available at https://www.usbr.gov/lc/region/g4000/NaturalFlow/current.html. Storage, regulated inflow, release, and evaporation data for Lake Powell and storage data for Lake Mead is available at https://www.usbr.gov/uc/water/hydrodata/reservoir_data/site_map.html. Observed streamflow data at the Colorado River above Diamond Creek gauge (USGS site number 09404200) can be found at https://waterdata.usgs.gov/monitoring-location/09404200. Hydropower generation data can be found at https://www.eia.gov/beta/electricity/data/browser.

### Code availability

The code used in this study is available at: https://github.com/bowenwang23/Colorado_River_WBM.git.

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

## Acknowledgements

We would like to acknowledge funding support for this project from the Department of Commerce NOAA project "Towards Predicting Drought and Subsequent Water Resource Challenges at Landscape-resolving Scales Across the Western U.S." (award no. NA23OAR4310633; B.W., B.B, S.R., and L.H.), the Department of Energy HyperFACETS project (award no. DE-SC0016605; B.B., A.H., S.R., and L.H.), and State of California support under California Energy Commission project "Development of Climate Projections for California and Identification of Priority Projections" (award no. EPC-20-006; B.B., A.H., and S.R.). We also want to thank Harrison B. Zeff and the Bureau of Reclamation for the opportunity to present our work to them and for providing feedback and their hydrologic ensemble for comparison. We also thank the computational support through the NCAR Computational and Information Systems Laboratory (CISL).

## Author contributions

B.W. led the development of WBM, data analysis, and manuscript writing. B.B. developed the initial project concept, developed the hydrologic dataset and guided the development of the project. A.H. provided general feedback and guidance. S.R. and L.H. developed and managed the forcing datasets. All authors reviewed and provided feedback to revisions of the manuscript.

## Competing interests

The authors declare no competing interests.
