## [Transparent Peer Review file · Nature Communications]

Disentangling Climate and Policy Uncertainties for the Colorado River Post-2026 Operations

Corresponding Author: Mr Bowen Wang

Version 0:

Reviewer comments:

Reviewer #1

(Remarks to the Author)

General comments regarding B. Wang et al manuscript

This paper presents the results of a modeling effort exploring different strategies for limiting consumptive uses in the Colorado River basin, with the primary metric being water storage in Lake Mead and Lake Powell, where most of the basin's water storage occurs. The presumption is that consumption must match an ever-declining water supply provided by natural runoff. B. Wang et al (subject manuscript) also address how well various strategies perform at the end of the 21st century, something addressed by Wheeler et al (2021) but not by Reclamation. The subject manuscript is like that undertaken by Wheeler et al (2021, 2022) and now being undertaken by Reclamation in its post-2026 effort. The earlier study, and Reclamation's on-going study, use different estimations of future climate and hydrology and different rules by which water consumption is reduced as inflow and reservoir storage decline.

B. Wang et al offer their modeling results without adequately comparing their findings with those of Wheeler et al (2021, 2022) or with the preliminary modeling presentations of Reclamation. Assuming the subject manuscript is eventually published, the paper is likely to become available at the same time that Reclamation will be releasing its own findings about the same subject. Readers will ask, "what is new here?" B. Wang et al would do well to anticipate this question.

Other comments

It is important that the hydrologic scenario developed by B. Wang et al is compared with the hydrologic scenarios developed and used by Reclamation. The authors should explain why their "dynamically downscaled, bias-corrected regional climate simulations based on 10 CMIP6 GCMs" are a better, or more acceptable, alternative approach to the scenario approach used by Reclamation in its on-going work. The paragraph (lines 629-642) in the methods presents this kind of information and should be shifted into the main body of the paper. B. Wang et al state that their hydrologic scenario is "a possible future hydrology drier than any of the Reclamation ensembles," and we assume their scenario is also drier than used by Wheeler et al (2022). How much drier? Is this drier scenario one that Reclamation should also be considering? B. Wang et al should also make specific comments about how their hydrologic scenario treats rare, extremely wet years and strings of very dry years. Those two extremes – wet and dry – especially stress reservoir management.

The alternative management scenarios evaluated by B. Wang et al are somewhat different than those evaluated by Wheeler et al (2021, 2022) and reflect recent proposals by the Upper Basin and Lower Basin on how water use should be reduced. The present paper uses a simpler water resource model than that used by Wheeler et al (2021, 2022) or by Reclamation in its Cloud-based web tool. Nevertheless, the findings of the present paper reach similar conclusions to those of Wheeler et al (2021, 2022)

- There isn't enough water supply to fully meet the growth projections of the Upper Basin and to meet the present uses of the Lower Basin and Mexico, and

- that present water consumption policy inadequately stabilizes reservoir storage in the mid- and later 21st century, assuming that climate continues to warm and runoff decrease.

The question is, what is new in the B. Wang et al study?

- a different hydrologic scenario drives the model
- different management alternatives are explored
- a simple water management model is used to explore alternative management strategies.

It would be helpful if the methodology of this paper explained why the model captures the essence of the management problem without using the detail and complexity of CRSS or CRMMS model tools used by the other studies or used in Reclamation's web-based model tool. If the present study's authors think that their approach is simpler and adequately addresses the major problems of the basin, then they should make that case. It would be helpful if the methodology explained if the paper's use of probabilistic methods that sample internal variability and use of the DMDU approach are unique and novel approaches not used by Wheeler et al (2022) nor by Reclamation in any of its on-going efforts.

It should be acknowledged that the models used in the present study accept that Upper Basin use will continue to grow as per the UCRC's 2016 aspirational growth projections. Hopefully, B. Wang et al realize that Reclamation is modeling the problem using a detailed approach that accounts for "hydrologic shortage" in the Upper Basin, i. e., the fact that UB demands cannot be met in dry years. In general, Reclamation assumes the magnitude of this shortage averages ~1.5 maf. If B. Wang et al are assuming that UB demands will be fully met in all years, then they are assuming a greater UB use than anyone else assumes. We are not sure that your assumption is fatal, but B. Wang et al should acknowledge that use of the 2016 Upper Basin demand projection is unlikely to be accepted in final negotiations for how to manage the future.

The differences between the B. Wang et al study and others nevertheless yield the same general conclusion – there isn't enough water and all of the alternatives still fail to achieve sustainability in the distant 21st century. Thus, the challenge of this paper is to explain why the water resource model used here is a better, or more realistic, model to use, or is computationally easier to use. The challenge is to explain why the hydrologic scenario used in this manuscript is more appropriate to consider the future climate.

Additional comments regarding the reservoir operations model

The authors addressed most of our concerns regarding modeling that we expressed in an earlier review. The current WBM model captures the trends and variability with some accuracy. The authors did a good job in testing 4 policies. However, there are still some unclear modelling components. We encourage the authors to make some further changes so that the model could be more robust and reliable.

- Hydropower generation should be a function of both water head and penstock releases. However, in Supplementary Figure 9, hydropower generation appears to only relate to water elevations. Given the same start elevation, hydropower generation will differ in relation to the amount of water released through the penstocks.
- It is not entirely clear how Lake Powell releases are characterized. In pdf line 509 and 510 (line numbers are not consistent with the word document), table S2 should be replaced by table S1. Other description about Lake Powell release policies proposed by UB or LB leads to "BoR, 2024", which is not in the Reference section. In the description from line 511 to 512, each policy decides the release volume based on the present-year storage and the predicted next-year storage. In the simulation, how does the manuscript predict next-year storage? We think authors can do a better job in explaining Lake Powell releases.
- Do the authors consider the limitations on release capacity for Glen Canyon Dam below its minimum power pool? When Lake Powell elevation goes below minimum power pool (3490 feet), water can only be released through river outlets. River outlet capacity decreases as elevation goes down. For instance, when Lake Powell elevation is at 3400 feet, the outlet capacity is 4800 ft³/s (about 3.48 maf/year). That means Glen Canyon Dam cannot release more than this amount of water even if the policy requires greater releases. There are some probabilities when Lake Powell elevation goes to the dead pool. The river outlet capacity below, from white-paper-1.pdf is provided for your review.
- Please note that in the official models (CRSS or CRMMS) Reclamation used, bank storage is considered in the water balance equation. That's 6% to 8% of the changing reservoir storage. If WBM wants to capture more reality, this component needs to be considered, otherwise, authors can explain why ignoring this component.
- The citation in pdf line 105 "BoR, 2024" is not in the References section.

(Remarks on code availability)

Reviewer #2

(Remarks to the Author)

I am a previous reviewer of this manuscript, so I am largely judging this paper based off the previous response and the new additions. My earlier review mentioned the simplicity of some of the methodologies. The authors have now addressed these simplicities or at least justified their reasons. I am a hydrologist and am not familiar with all the minutiae and intricacies of Colorado River Basin policies and laws. Thus, I am reviewing this more from a hydroclimatological viewpoint.

First, I want to commend the authors for doing a thorough job at addressing many of my concerns. My issues on this version

of the are largely minor, with many of my major issues being addressed in the previous revision.

[1] In the abstract for the phrase “at least 80% likelihood of reach dead pool at least once”, does this dead pool refer to both reservoirs reaching dead pool or only one of the two reservoirs reaching dead pool? Not clear on this.

[2] In the abstract, I think it would benefit the reader to know why these reservoirs are reaching dead pool? I know it's a decline in runoff, but is that due to a decrease in precip? Just one sentence or phrase would go a long way.

[3] In the intro in the paragraph starting with “Understanding how climate...” the authors should also mention the uncertainty of hydrologic models that are being used in these studies. Along with the other uncertainties, this is an additional important layer of uncertainty. There has been quite a bit of work on this, but one is Clark et al. (2016) – see below. I think it's important for the authors to mention that the hydrologic model that they are using has assumptions; other hydrologic models have different assumptions. Both may give you different results. I am not suggesting that the authors need to add an additional model, but perhaps mention that it's possible that the conclusions are an artifact of the model.

Clark, M. P., Wilby, R. L., Gutmann, E. D., Vano, J. A., Gangopadhyay, S., Wood, A. W., ... & Brekke, L. D. (2016). Characterizing uncertainty of the hydrologic impacts of climate change. *Current climate change reports*, 2, 55-64.

[4] It's not clear. Did the authors use an annual time step (Jan – Dec) or a water year time step (Oct to Sept)? The authors mention annual time step, but just wanted to ensure that this is correct. Most (?) hydrologic work (especially in the West) is based on the water year to capture snow processes. If the authors used the annual time step, does this have any influence of the results?

[5] Figure S6 needs a better description, either within the text or in the caption. I do not understand why the red line is so high. Is the future UB demand biased?

[6] Figure 1 needs the subpanel letter labeling.

[7] In the “Future Reservoir...” section, I think the authors need to make it clear why they used the Monte Carlo approach and not just the 10 GCMs, as most studies solely use GCM outputs. Is this solely for the probabilistic approach discussed later on in the paragraph? Additionally, the authors mention 1,000 realizations. Is that the # of GCMs *1000, so 10,000 in total?

(Remarks on code availability)

Reviewer #3

(Remarks to the Author)

(Remarks on code availability)

Reviewer #4

(Remarks to the Author)

The authors have responded to some of my previous comments, and the paper is certainly improved.

“All policies can exhibit tipping points where the reservoir levels can change rapidly with only a slight change in natural flow. We argue that a robust policy should buffer the reservoir from such sudden changes under all conditions, and yield reservoir storage and water delivery outcomes consistent with changes in hydrologic condition”
This is MUCH better than the previous hokum. Nice.

Evaluating societal impact – this is still a misnomer. Why pretend you are saying anything (direct) about societal impact when you are just evaluating water supply and hydropower? The added caveats are OK, but lipstick on a pig. The urge to oversell is strong in this one, paduwan. What you have done is pretty good, so just say those things.

I notice the following in the first and second versions. From:

“We find that water levels Mead and Powell reached during the megadrought will likely become the norm by 2065, and both reservoirs have a greater than 65% likelihood of reaching dead pool by the end of the 21st century. There is also over 30% probability that under existing policy, neither Mead nor Powell can generate any hydroelectricity between 2080 and 2100.”

To:

“Under existing policy, the reservoirs will face substantial risks before 2060, with at least 80% likelihood of reaching dead pool at least once.”

Will we get to 90% if you rerun it again ;) In all seriousness, what changed in the analysis?

At 98. This paragraph is pretty good, but a clearer description with just a touch more detail on the differences among the scenarios, perhaps adding some rationale or stated intent, would be helpful.

There are many tradeoffs in the policies, which your data seem to shed some potentially useful light on, but this is buried in some dense text. If this could be turned into a table that lays this out clearly it would be much better, helping to articulate who bears the costs and risks in each scenario, perhaps pro/con type framing. That could turn this into a useful paper.

The tipping point discussion is improved, but still falls short. It would seem the authors are well placed to recommend modifications to existing policies, perhaps conditional operations rules, that could avoid these events. Can you run anti-tipping point policy scenarios and suggest what those might entail, including their costs in terms of water supply and hydropower? That would square the circle nicely whereas now the tipping points are presented as sort of a bugbear or vague threat.

Overall, the paper is getting closer to talking directly about policies. I believe it could take a step further as it sort of dances around the implications of the analysis. Can the authors get more explicit about things like – the UB proposal benefits UB states in X ways, and carries Y risks for the entire system, and correspondingly for the other policy proposals? This discussion could be more direct I would think, as much of it is left to the reader to puzzle out.

For example: “We argue that a robust alternative policy should serve as a buffer to sudden changes in reservoir conditions and yield reservoir storage and water delivery outcomes consistent with the hydrologic conditions.” Wait, which one are you talking about here? Are any of the three alternatives equally good, or are you referring to something hidden or as yet to be developed?

With revisions the paper could be publishable and even useful.

(Remarks on code availability)

Version 1:

Reviewer comments:

Reviewer #1

(Remarks to the Author)

Manuscript provides a well written context of how this work (model and results) fits into the context of on-going negotiations of how to manage the Colorado River. I appreciate the authors efforts to provide this context in their introduction. The most important contribution of this paper is offering a simple model with which to examine policy options. no one knows what the ultimate agreement will be, and it surely will not be one of the alternatives analyzed in this paper. that is fine. the point is to provide a means to evaluate alternatives. similarly, no one knows what future Upper Basin runoff will be (although I think that this paper does not consider sufficiently dry scenarios), but you might be providing tools to use other hydrologies. I think that it is significant that all the alternatives include some risk of going to dead pool. if such were to come to pass, UB use would be drastically cut. who would ever keep irrigating alfalfa for export when water supplies in major cities of the LB are threatened? I think that your point is that none of these alternatives eliminates what is an unacceptable risk; sustainable proposals wil have to include larger cuts, especially in the UB where the economic value of water is much less than in the LB.

(Remarks on code availability)

REVIEWER COMMENTS

We appreciate the time and effort from all reviewers for reviewing this manuscript and providing constructive feedback. We would like to point out that since we last submitted this paper (September 2024) and likely after all reviews were submitted, in January 2025, the Bureau of Reclamation published a report named “Alternatives Report: Post-2026 Operational Guidelines and Strategies for Lake Powell and Lake Mead,” which outlines 5 alternative policies that Reclamation is now evaluating. While our analysis was done before this report came out and was thus based on alternatives published earlier (March 2024), the alternatives we considered here reflect variations to existing policy that are generally considered for post-2026 operations, and, to our knowledge, are nonetheless the most up-to-date alternatives that have been evaluated in the scientific literature.

Reviewer #1 (Remarks to the Author):

General comments regarding B. Wang et al manuscript

This paper presents the results of a modeling effort exploring different strategies for limiting consumptive uses in the Colorado River basin, with the primary metric being water storage in Lake Mead and Lake Powell, where most of the basin’s water storage occurs. The presumption is that consumption must match an ever-declining water supply provided by natural runoff. B. Wang et al (subject manuscript) also address how well various strategies perform at the end of the 21st century, something addressed by Wheeler et al (2021) but not by Reclamation. The subject manuscript is like that undertaken by Wheeler et al (2021, 2022) and now being undertaken by Reclamation in its post-2026 effort. The earlier study, and Reclamation’s ongoing study, use different estimations of future climate and hydrology and different rules by which water consumption is reduced as inflow and reservoir storage decline.

B. Wang et al offer their modeling results without adequately comparing their findings with those of Wheeler et al (2021, 2022) or with the preliminary modeling presentations of Reclamation. Assuming the subject manuscript is eventually published, the paper is likely to become available at the same time that Reclamation will be releasing its own findings about the same subject. Readers will ask, “what is new here?” B. Wang et al would do well to anticipate this question.

Thank you for the feedback and we appreciate your time reviewing this paper again. We added a paragraph at the end of the Introduction section to explicitly summarize the relevance/insight provided by our study relative to Wheeler and Reclamation studies. Note, earlier in this section we provide details on each of these new contributions that we provide (including the CMIP6 modeling workflow, the WBM, the evaluation of alternatives not previously assessed, and the

unique methods for sampling and removing internal variability that collectively provide robust analysis and insights of policies and reservoir response in a manner not done previously).

Other comments

It is important that the hydrologic scenario developed by B. Wang et al is compared with the hydrologic scenarios developed and used by Reclamation.

Figure S14 compares the hydrologic scenarios we use to the BoR scenarios.

The authors should explain why their “dynamically downscaled, bias-corrected regional climate simulations based on 10 CMIP6 GCMs” are a better, or more acceptable, alternative approach to the scenario approach used by Reclamation in its on-going work. The paragraph (lines 629-642) in the methods presents this kind of information and should be shifted into the main body of the paper. B. Wang et al state that their hydrologic scenario is “a possible future hydrology drier than any of the Reclamation ensembles,” and we assume their scenario is also drier than used by Wheeler et al (2022). How much drier? Is this drier scenario one that Reclamation should also be considering?

In the updated introduction, we point to the fact that we are using high-resolution, dynamically downscaled CMIP6 data (Line 98-109). Given space limitations, please refer to the discussion in the methods, with reference to Figure S14, which outlines that our ensemble is slightly drier compared to CMIP3 and CMIP5 given our use of the latest generation of CMIP data (CMIP6). Wheeler et al. (2022) sampled natural flow from the megadrought. Wheeler included the years 2000-2018, which had a mean flow of 12.4 MAF/year, which is drier than the average conditions considered by us as well as the Reclamation.

B. Wang et al should also make specific comments about how their hydrologic scenario treats rare, extremely wet years and strings of very dry years. Those two extremes – wet and dry – especially stress reservoir management.

The WBM as outlined in the manuscript and methods explicitly accounts for annual natural flow in the UCRB. The time-series of typical, wet, dry, strings of dry/wet conditions from each downscaled GCM is explicitly represented. The internal variability sampling as described in the methods also accounts for autocorrelation to the previous year's natural flow.

The alternative management scenarios evaluated by B. Wang et al are somewhat different than those evaluated by Wheeler et al (2021, 2022) and reflect recent proposals by the Upper Basin and Lower Basin on how water use should be reduced. The present paper uses a simpler water resource model than that used by Wheeler et al (2021, 2022) or by Reclamation in its Cloud-based web tool. Nevertheless, the findings of the present paper reach similar conclusions to those of Wheeler et al (2021, 2022)

- There isn't enough water supply to fully meet the growth projections of the Upper Basin and to meet the present uses of the Lower Basin and Mexico, and
- that present water consumption policy inadequately stabilizes reservoir storage in the mid- and later 21st century, assuming that climate continues to warm and runoff decrease.

The question is, what is new in the B. Wang et al study?

- a different hydrologic scenario drives the model
- different management alternatives are explored
- a simple water management model is used to explore alternative management strategies.

As mentioned in response to your other comments above and below, the Introduction was updated to explicitly highlight these bullet points and the value of our analysis (Line 81-90, 98-109, 146-157). To summarize, our simple WBM is a rapid and flexible tool that allows us to perform the sampling methods we employed (one that is probabilistic by introducing internal variability and one that is based on a range of plausible average flows by effectively removing internal variability – the latter leads to equilibrium conditions for a given natural flow). These sampling methods have not been used by Wheeler et al. or the Bureau of Reclamation, and they are based on the latest phase of dynamically downscaled CMIP data. Results from our study thus provide two unique perspectives (probabilistic vs equilibrium outcome) on the performance of alternatives with respect to reservoir storage and resulting water supply and hydropower.

It would be helpful if the methodology of this paper explained why the model captures the essence of the management problem without using the detail and complexity of CRSS or CRMMS model tools used by the other studies or used in Reclamation's web-based model tool. If the present study's authors think that their approach is simpler and adequately addresses the major problems of the basin, then they should make that case. It would be helpful if the methodology explained if the paper's use of probabilistic methods that sample internal variability and use of the DMDU approach are unique and novel approaches not used by Wheeler et al (2022) nor by Reclamation in any of its on-going efforts.

We added a statement in the Introduction (Line 81-90) that our WBM is simpler than CRSS or CRMMS but accurately represents the long-term changes in the storage at Powell and Mead, which is what we are interested in for this study to understand the climate-change-driven risks at the reservoirs. We also point out that the methods we use are unique from that used by Wheeler and the Bureau of Reclamation, and that the flexible and rapid WBM which does not require proprietary software allows us to perform the sampling methods that we employ. Given space limitations, we leave the description of why our model works to the methods, and added the following statement (Lines 669-678) to address why our model captures the overall conditions that CRSS represents in much greater detail:

“Our WBM is beneficial to our study because it is 1) free (python-based), 2) computationally efficient, and 3) flexible with different policy alternatives.. The Colorado River Simulation System (CRSS) requires a RiverWare license, and while includes finer details like the diversions and smaller reservoirs in the CRB, its longer run-times make it infeasible to simulate over a large number of time series for long time-periods. In contrast, while our model does not explicitly represent these minor systems, it captures the cumulative impact on the total volume of water reaching Lakes Powell and Mead and significantly reduces the amount of time needed for each simulation. Similarly, although CRSS can model local conservation efforts, our approach focuses on the aggregate effects of conservation or demand across the upper and lower basin. Further details on CRSS can be found in Bureau of Reclamation (2007).”

It should be acknowledged that the models used in the present study accept that Upper Basin use will continue to grow as per the UCRC’s 2016 aspirational growth projections. Hopefully, B. Wang et al realize that Reclamation is modeling the problem using a detailed approach that accounts for “hydrologic shortage” in the Upper Basin, i. e., the fact that UB demands cannot be met in dry years. In general, Reclamation assumes the magnitude of this shortage averages ~1.5 maf. If B. Wang et al are assuming that UB demands will be fully met in all years, then they are assuming a greater UB use than anyone else assumes. We are not sure that your assumption is fatal, but B. Wang et al should acknowledge that use of the 2016 Upper Basin demand projection is unlikely to be accepted in final negotiations for how to manage the future.

Thank you for the feedback. We are indeed aware that the full UB water demand (like the ones stated in the UCRC 2016 projections) are unlikely to be fully realized. We have noted in the manuscript that the projections in this study are not based on the raw UCRC projections, but an adjusted version that matches well with the observed UB depletion patterns, which is roughly 1 MAF less than the UCRC 2016 projections at any given year (**Fig S6**). We are also aware that depending on the policy, even this adjusted UB demand projection may not be fully fulfilled in the final policy, for which we added a statement in the Introduction (Line 126-135) to more clearly acknowledge.

The differences between the B. Wang et al study and others nevertheless yield the same general conclusion – there isn’t enough water and all of the alternatives still fail to achieve sustainability in the distant 21st century. Thus, the challenge of this paper is to explain why the water resource model used here is a better, or more realistic, model to use, or is computationally easier to use. The challenge is to explain why the hydrologic scenario used in this manuscript is more appropriate to consider the future climate.

As we explained in the earlier response, we added an explanation (Line 84-90) that our model is sufficient to address the question of long-term reservoir changes in the CRB and is also more computationally efficient than CRSS and CRMMS. We also added an explanation of the unique

values our hydrologic scenarios bring to understanding the future hydrology of the CRB (Line 98-105). Specifically, 1) the CMIP6 models have been shown to better represent the hydroclimate of the UCRB than the CMIP3 and CMIP5 models used by Reclamation, 2) the dynamical downscaling and hydrologic modeling approaches we use - as opposed to the statistical downscaling approach used by Reclamation - provide a more realistic, high-resolution, physics-based projection of the nonstationary hydrologic changes in the UCRB, and 3) the hydrologic model we use has been calibrated and well validated to the UCRB (**Figure S3a**), while the CMIP3 and CMIP5 hydrologic traces used by Reclamation are modeled by an uncalibrated hydrologic model (BoR, 2012). It has been shown that calibration can be critical to the accuracy of hydrologic simulations in the western US (Bass et al., 2023; Su et al., 2024).

Additional comments regarding the reservoir operations model

The authors addressed most of our concerns regarding modeling that we expressed in an earlier review. The current WBM model captures the trends and variability with some accuracy. The authors did a good job in testing 4 policies. However, there are still some unclear modelling components. We encourage the authors to make some further changes so that the model could be more robust and reliable.

- Hydropower generation should be a function of both water head and penstock releases. However, in Supplementary Figure 9, hydropower generation appears to only relate to water elevations. Given the same start elevation, hydropower generation will differ in relation to the amount of water released through the penstocks.

Thank you for the suggestion. We updated the linear regression such that the predictor for hydropower generated is the water level multiplied by annual total release. Whenever the water level is below 3490 ft, the hydropower generation is set to 0. This regression very closely matches the observations, as shown in the updated **Fig. S9**. The results have been updated accordingly in the manuscript.

- It is not entirely clear how Lake Powell releases are characterized. In pdf line 509 and 510 (line numbers are not consistent with the word document), table S2 should be replaced by table S1. Other description about Lake Powell release policies proposed by UB or LB leads to “BoR, 2024”, which is not in the Reference section. In the description from line 511 to 512, each policy decides the release volume based on the present-year storage and the predicted next-year storage. In the simulation, how does the manuscript predict next-year storage? We think authors can do a better job in explaining Lake Powell releases.

We’d like to clarify that the captions for **Tables S1-3** are above the respective tables, not below, and the reference in the paragraph “The release from Powell each year...” in the Water Budget Model subsection of the Methods section was mislabeled. We also provided a more elaborate explanation on the calculations for Powell release in the same paragraph that you referred to.

- Do the authors consider the limitations on release capacity for Glen Canyon Dam below its minimum power pool? When Lake Powell elevation goes below minimum power pool (3490 feet), water can only be released through river outlets. River outlet capacity decreases as elevation goes down. For instance, when Lake Powell elevation is at 3400 feet, the outlet capacity is 4800 ft³/s (about 3.48 maf/year). That means Glen Canyon Dam cannot release more than this amount of water even if the policy requires greater releases. There are some probabilities when Lake Powell elevation goes to the dead pool. The river outlet capacity below, from white-paper-1.pdf is provided for your review.

Thank you for pointing this out. Based on the Schmidt et al. (2016) white paper, we incorporated an upper limit on Lake Powell release based on Powell's current year storage. The figures and results in the manuscript have been updated accordingly. In general, incorporating this change makes it more likely that each reservoir will reach dead and/or inactive at least once given a time series of projected streamflow (**Fig. 2c-f**), but reduces the amount of time that it is under such conditions (**Fig. 2a-b**).

- Please note that in the official models (CRSS or CRMMS) Reclamation used, bank storage is considered in the water balance equation. That's 6% to 8% of the changing reservoir storage. If WBM wants to capture more reality, this component needs to be considered, otherwise, authors can explain why ignoring this component.

Thank you for the suggestion. We followed the CRSS approach that represented the amount of water entering or leaving bank storage each year by a fixed percent of reservoir storage change that year. We compared the model projections with and without consideration of bank storage using the 10 dynamically downscaled streamflow time series (not the Monte Carlo ensemble). The figure on the left shows the model projections without considering bank storage, and on the right shows bank storage as 8% of reservoir change, and the results are clearly very similar. Thus, we find it unnecessary to include bank storage as an additional component in the water budget.

• The citation in pdf line 105 “BoR, 2024” is not in the References section.

We added the citation for this reference. It is referring to the letters from the Lower and Upper Basin states that explain their proposals.

Reviewer #2 (Remarks to the Author):

I am a previous reviewer of this manuscript, so I am largely judging this paper based off the previous response and the new additions. My earlier review mentioned the simplicity of some of the methodologies. The authors have now addressed these simplicities or at least justified their reasons. I am a hydrologist and am not familiar with all the minutiae and intricacies of Colorado River Basin policies and laws. Thus, I am reviewing this more from a hydroclimatological viewpoint.

First, I want to commend the authors for doing a thorough job at addressing many of my concerns. My issues on this version of the are largely minor, with many of my major issues being addressed in the previous revision.

Thank you again for the feedback and your time reviewing this paper. We appreciate your comments and suggestions in both rounds of review.

[1] In the abstract for the phrase “at least 80% likelihood of reach dead pool at least once”, does this dead pool refer to both reservoirs reaching dead pool or only one of the two reservoirs reaching dead pool? Not clear on this.

We clarified in the abstract that the original sentence meant the likelihood of each reservoir individually reaching the dead pool.

[2] In the abstract, I think it would benefit the reader to know why these reservoirs are reaching dead pool? I know it's a decline in runoff, but is that due to a decrease in precip? Just one sentence or phrase would go a long way.

We have modified the abstract to include that the decline in runoff is driven by a warming-induced increase in evapotranspiration, rather than a decline in precipitation, as we have shown in Figure 1.

[3] In the intro in the paragraph starting with "Understanding how climate..." the authors should also mention the uncertainty of hydrologic models that are being used in these studies. Along with the other uncertainties, this is an additional important layer of uncertainty. There has been quite a bit of work on this, but one is Clark et al. (2016) – see below. I think it's important for the authors to mention that the hydrologic model that they are using has assumptions; other hydrologic models have different assumptions. Both may give you different results. I am not suggesting that the authors need to add an additional model, but perhaps mention that it's possible that the conclusions are an artifact of the model.

Clark, M. P., Wilby, R. L., Gutmann, E. D., Vano, J. A., Gangopadhyay, S., Wood, A. W., ... & Brekke, L. D. (2016). Characterizing uncertainty of the hydrologic impacts of climate change. *Current climate change reports*, 2, 55-64.

Thank you for the suggestion. In introduction we added this point and citation to Clark et al. 2016: "Furthermore, the sensitivity of runoff to changes in temperature and precipitation remains uncertain from observations and unique hydrology models (Clark et al. 2016; Udall and Overpeck, 2017; Hoerling et al., 2019; Lehner et al., 2019; Milly and Dunne, 2022)...". Also, In Introduction we added a general statement and citation regarding the sensitivity to modeling choices: "Our analysis relies on a specific hydroclimate modeling workflow, which includes GCM selection, downscaling, bias correction, hydrologic modeling, and water budget modeling (**Methods**). Although unique modeling choices at each of these steps can introduce additional levels of uncertainty (e.g. Bosshard et al. 2013), our choices were carefully considered and validated."

[4] It's not clear. Did the authors use an annual time step (Jan – Dec) or a water year time step (Oct to Sept)? The authors mention annual time step, but just wanted to ensure that this is correct. Most (?) hydrologic work (especially in the West) is based on the water year to capture snow processes. If the authors used the annual time step, does this have any influence of the results?

The model is based on calendar year time steps because it is the time unit of water accounting for all the water deliveries/consumptions in the Upper and Lower Basin states. For this reason, it is difficult to fully adjust the model to function on water years. If we run the model using runoff by water year and water deliveries by calendar year, the results are almost identical. This information is now provided in the Methods sections (Line 543-544).

[5] Figure S6 needs a better description, either within the text or in the caption. I do not understand why the red line is so high. Is the future UB demand biased?

The red line is the data that we directly extract from the Upper Colorado River Commission (UCRC) projections. The blue line is what we find to be a more realistic estimate based on historical consumptive use data and is what we use in this paper. This information is also provided in the caption of the figure.

[6] Figure 1 needs the subpanel letter labeling.

Thank you for pointing this out. We added the letter labels for each subpanel.

[7] In the “Future Reservoir...” section, I think the authors need to make it clear why they used the Monte Carlo approach and not just the 10 GCMs, as most studies solely use GCM outputs. Is this solely for the probabilistic approach discussed later on in the paragraph? Additionally, the authors mention 1,000 realizations. Is that the # of GCMs *1000, so 10,000 in total?

Thank you for your questions. We have demonstrated in the previous section that within our 10-GCM ensemble (which is smaller than a typical GCM ensemble of either raw or statistically downscaled GCMs), there is considerable uncertainty with trends in precipitation due to the large influence of internal variability, which influences the spread in runoff projections. The Monte Carlo approach is designed to create a much larger ensemble that captures internal variability which is poorly sampled by 10 GCMs on their own, and this provides us with more robust statistics on the likelihood a given reservoir condition may occur. Yes, we produce 1000 realizations for each GCM, thus 10,000 simulations in total.

Reviewer #3 (Remarks to the Author):

Thank you for your time reviewing this manuscript.

Reviewer #4 (Remarks to the Author):

The authors have responded to some of my previous comments, and the paper is certainly improved.

We appreciate your constructive feedback on the manuscript and for your time reviewing this paper again. We are glad that you find the manuscript improved from the previous version.

“All policies can exhibit tipping points where the reservoir levels can change rapidly with only a slight change in natural flow. We argue that a robust policy should buffer the reservoir from such sudden changes under all conditions, and yield reservoir storage and water delivery outcomes consistent with changes in hydrologic condition”

This is MUCH better than the previous hokum. Nice.

We appreciate your previous suggestions on the analysis for the tipping points, and we agree that they helped improve the manuscript.

Evaluating societal impact – this is still a misnomer. Why pretend you are saying anything (direct) about societal impact when you are just evaluating water supply and hydropower? The added caveats are OK, but lipstick on a pig. The urge to oversell is strong in this one, paduwan. What you have done is pretty good, so just say those things.

Thank you for your positive feedback on the analysis. We updated the section header to “Evaluating Impacts on Water Supply and Hydropower Generation”.

I notice the following in the first and second versions. From:

“We find that water levels Mead and Powell reached during the megadrought will likely become the norm by 2065, and both reservoirs have a greater than 65% likelihood of reaching dead pool by the end of the 21st century. There is also over 30% probability that under existing policy, neither Mead nor Powell can generate any hydroelectricity between 2080 and 2100.”

To:

“Under existing policy, the reservoirs will face substantial risks before 2060, with at least 80% likelihood of reaching dead pool at least once.”

Will we get to 90% if you rerun it again ;) In all seriousness, what changed in the analysis?

We appreciate the careful observation. The modeled probabilities have changed since the model went through substantial revisions since the last round of review. The most important change that led to the increase in projected risks was an explicit representation of Upper Basin depletions in the model based on projected values in **Figure S6**. Without this component, the model was implicitly assuming an UB depletion that equals to the historical average of 3.5 MAF/year, which was much lower than what would be realistic. The representation of projected UB demand was incorporated based on previous reviewer comments. You will also notice that the numbers changed again in this revised version of the manuscript. This is because reviewer #1 pointed out

that we neglected the physical limits on Powell release when Powell's water level is low. In general, incorporating this change makes it more likely that each reservoir will reach dead and/or inactive at least once given a time series of projected streamflow (**Fig. 2c-f**), but reduces the amount of time that it is under such conditions (**Fig. 2a-b**). After these revisions, we believe the model in its current form has comprehensively accounted for the major processes related to the reservoir conditions in the basin.

At 98. This paragraph is pretty good, but a clearer description with just a touch more detail on the differences among the scenarios, perhaps adding some rationale or stated intent, would be helpful.

Following your suggestion, we provided a more elaborate discussion about the UB and LB proposals in this paragraph.

There are many tradeoffs in the policies, which your data seem to shed some potentially useful light on, but this is buried in some dense text. If this could be turned into a table that lays this out clearly it would be much better, helping to articulate who bears the costs and risks in each scenario, perhaps pro/con type framing. That could turn this into a useful paper.

Thank you for the suggestions. The tradeoffs among healthy reservoir storage, water delivery, and hydropower generation are contingent on both the policy choice and the hydrologic conditions, as we show in Figure 4. It is difficult to concisely summarize this information in a table but easier to do so with words. So we added a concise summary of the effects of each policy in the Discussion section, especially on LB2024 and UB2024 which are more relevant for decision-making (Line 445-461).

The tipping point discussion is improved, but still falls short. It would seem the authors are well placed to recommend modifications to existing policies, perhaps conditional operations rules, that could avoid these events. Can you run anti-tipping point policy scenarios and suggest what those might entail, including their costs in terms of water supply and hydropower? That would square the circle nicely whereas now the tipping points are presented as sort of a bugbear or vague threat.

Following your suggestion, we provided a more explicit recommendation (Line 482-485) that a robust policy against tipping points would need to introduce delivery reduction schemes where reduction always increases when storage level decreases. However, we decided to not include recommendations of specific schemes of anti-tipping point policies since this is beyond the scope of this paper, which focuses on evaluating policies that have already been proposed.

Overall, the paper is getting closer to talking directly about policies. I believe it could take a step further as it sort of dances around the implications of the analysis. Can the authors get more explicit about things like – the UB proposal benefits UB states in X ways, and carries Y risks for

the entire system, and correspondingly for the other policy proposals? This discussion could be more direct I would think, as much of it is left to the reader to puzzle out.

Thank you for the suggestion. In the discussion section (Line 445-461), we added an explicit discussion of the consequences of adopting each policy and the tradeoffs involved.

For example: “We argue that a robust alternative policy should serve as a buffer to sudden changes in reservoir conditions and yield reservoir storage and water delivery outcomes consistent with the hydrologic conditions.” Wait, which one are you talking about here? Are any of the three alternatives equally good, or are you referring to something hidden or as yet to be developed?

This is referring to the discussion of the DMDU analysis and tipping points in the previous paragraph, where we argue that the reductions must reflect the reservoir conditions. This is not referring to a specific policy but a general statement based on the previous discussion on the equilibrium analysis.

With revisions the paper could be publishable and even useful.

We appreciate your feedback on the paper.

Specific comments:

Line 42 – The Law of the River allocates 16 maf in the United States – 8.5 maf in the LB and 7.5 maf in the UB, as well as 1.5 maf to Mexico (except in times of extraordinary drought). see Kuhn and Fleck (2019). Kuhn and Fleck (2019) is the definitive book describing how and why the Law of the River was negotiated and what was known about the river's hydrology.

Line 46 – The LB is fully allocated and the UB never will be fully developed. No one expects the UB to ever achieve its aspirational projections of use.

Line 49 – this paragraph implies that the Millennium Drought has ended, presumably because of the 2023 wet year. But we are right back to low runoff in 2024 and 2025. The general view is that the Millennium Drought is the new normal and that 2002-2004 and 2020-2022 were unusual episodes of very dry conditions. Perhaps you think differently but make clear whether you are asserting that the Millennium Drought has ended, or not. Reservoir storage reached its lowest in March 2023.

Line 83 – Lake Powell and Lake Mead are proper names. Both bodies of water are reservoirs and neither is a lake. thus, you should say “Lake Powell (hereafter Powell) and Lake Mead (hereafter Mead) reservoirs”.

Line 118 – should be “among,” not “between.”

Line 187 – Fig. 1(d) does not have a black dashed line.

Line 209 – point for discussion ... does the initial condition matter in the model results? does the model include the large recovery of storage in 2023? The falling storage in 2024 and 2025?

Line 259-260 – if there were another critical aspect, it would be annual release from Powell, which has significant impacts to ecosystem and recreational resources in Grand Canyon.

Line 294, line 382 – Average annual natural flow between 2000 and 2024 is 12.4 maf/yr. it appears that you assume natural flow is more than this. it seems to me that your assumptions about future hydrology are not sufficiently dry. You might want to discuss in Discussion.